# Health octo tool matches personalized health with rate of aging

Sh Salimi[1] ✉, A. Vehtari [2], M. Salive [3], M. Kaeberlein [4], D. Raftery [5] & L. Ferrucci [6]

Medical practice mainly addresses single diseases, neglecting multimorbidity as a heterogeneous health decline across organ systems. Aging is a multidimensional process and cannot be captured by a single metric. Therefore, we assessed global health in longitudinal studies, BLSA ($n = 907$), InCHIANTI ($n = 986$), and NHANES ($n = 40,790$), by examining disease severities in 13 bodily systems, generating the Body Organ Disease Number (BODN), reflecting progressive system morbidities. We used Bayesian ordinal models, regressing BODN over organ specific and all organs disease severities to obtain Body System-Specific Clocks and the Body Clock, respectively. The Body Clock is BODN weighted by the posterior coefficient of diseases for each individual. It supersedes the frailty index, predicting disability, geriatric syndrome, SPPB, and mortality with ≥90% accuracy. The Health Octo Tool, derived from Bodily System-Specific Clocks, the Body Clock and Clocks that incorporate walking speed and disability and their aging rates, captures multidimensional aging heterogeneity across organs and individuals.

Current medical practice focuses primarily on the diagnosis and cure of single diseases and only rarely considers comorbidities, even for diseases that are highly prevalent in the population such as diabetes and hypertension. This focus on specific diseases may overlook the broader implications of multimorbidity on global health in older adults, where the coexistence of two or more diseases is not only highly prevalent but also an indicator of higher disease susceptibility due to entropic molecular and cellular damage not adequately counteracted by mechanisms of allostatic resilience[1–3]. Recently, it has been proposed that biological aging is the root cause of age-associated pathologies across multiple body systems, functional decline, and disability, through the accumulation of damage that surpasses the body's resilience and homeostatic mechanisms, leading to declining health and emerging clinically as multimorbidity[4,5]. Accordingly, multimorbidity, a strong clinical risk factor for disability and mortality[6], is often associated with polypharmacy and iatrogenesis and imposes significant burdens on individuals and society[7].

Despite the importance of multimorbidity and the growing evidence that it is not merely the sum of single diseases, there are currently no widely acknowledged measures of multimorbidity that fully account for its complexity and the connection between the rate of aging and the rising susceptibility to chronic diseases development. Guidelines for the clinical management of multimorbidity in older patients have only recently appeared in the literature[8,9]. However, diagnostic tools proposed in the literature focus on the number of chronic diseases[7,10]; the proportional number of deficits over the total number of measured deficits, as in the Frailty Index (FI)[11–16]; or components of FI to predict mortality using Machine Learning[17,18]. Other authors have proposed weighting the contribution of specific diseases based upon their impact on functional status[19,20] or mortality[21,22]. These tools overlook the possibility that combinations of chronic diseases or deficits can potentially be heterogeneous, and that organ system health can differentially impact whole-body health entropy and functional and disability status in different individuals. Current

[1]Department of Anesthesiology and Pain Medicine, University of Washington, Seattle, WA, USA. [2]Department of Computer Science, Aalto University, Aalto, Finland. [3]Division of Geriatrics and Clinical Gerontology, National Institute on Aging, Bethesda, MD, USA. [4]Optispan Inc, Seattle, WA, USA. [5]Department of Anesthesiology and Pain Medicine, University of Washington, Northwest Metabolomics Research Center, Seattle, WA, USA. [6]Intramural Research Program, National Institute on Aging, Baltimore, MD, USA. ✉e-mail: ssalimi2@uw.edu

approaches, for example, may underestimate the health burden in patients with one very severe disease or fail to recognize that individuals with multiple diseases can still maintain functional resilience or live long, autonomous lives. In fact, there is evidence that consistent dynamic compensatory mechanisms might counteract a specific organ system's dysfunction, while global physical and cognitive functions remain preserved despite the severity of a single system's deficit[19,23]. Therefore, focusing only on late-life outcomes for reference might skew the metric toward older chronological ages and underestimate the cumulative effect of health decline that has already occurred at an earlier stage[22,24,25]. In addition, indices developed to predict mortality are prone to validity issues as incidence rates and causes of mortality have varied over the decades and are likely to vary in the future[21]. Moreover, there is currently no metric that captures the rate of aging in terms of multimorbidity components. In light of these limitations, the development of novel metrics that account for early-onset diseases/ deficits, their severity, and the complexity of multimorbidity interpreted as entropic fluctuations of health independent of chronological age would be highly desirable[26,27].

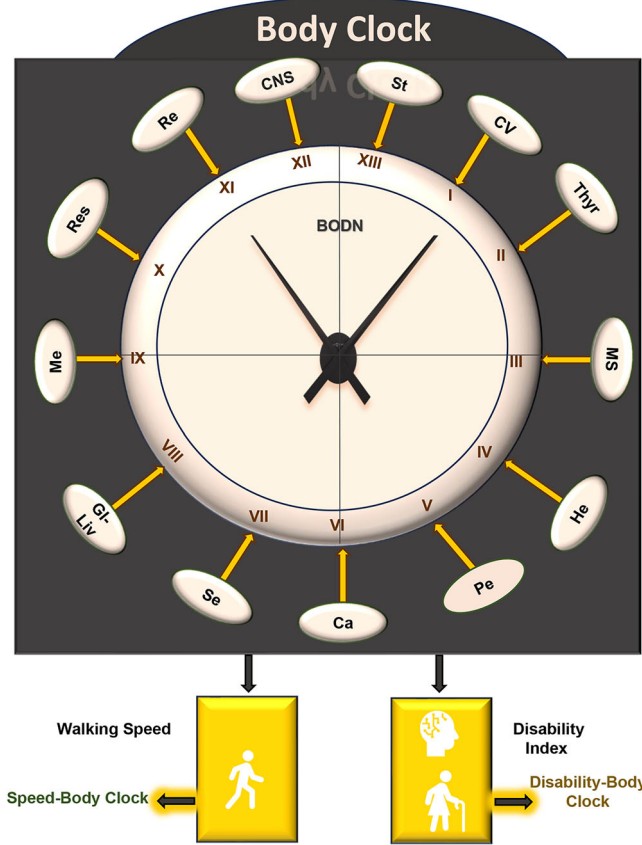

**Fig. 1 | The Health Octo Tool is comprised of eight components designed to assess multidimensional health.** The Clock components of the Health Octo Tool. Body Organ Disease Number (BODN) quantifies organ systems with at least one disease, considering diseases across multiple systems. Post hoc analyses of the organ-specific diseases predicting BODN yield the Bodily Organ-Specific Clock (BSC). Including all organ systems in the Bayesian ordinal regression generates the Body Clock. The Body Clock impacts walking speed, resulting in the Speed-Body Clock, and functional and cognitive disability, resulting in the Disability-Body Clock. System abbreviations: CNS Central Nervous System, St Stroke, CV Cardiovascular, Thyr Thyroid, MS Musculoskeletal, He Hematology, Pe Periodontal system, Ca Cancer, Se Sensory, GI-Liv Gastrointestinal and Liver system, Me Metabolic system, Res Respiratory system, Re Renal System. The interactive sunburst graphs shown at https://bodiagesystem.shinyapps.io/BODN_BLSA/ reveal increases in multisystem morbidity and BODN values with aging and heterogenous combinations of systems in the BLSA data. Sample size n = 907.

To address this challenge, we first developed the Body Organ Disease Number (BODN) as the number of organ systems with at least one deviation from health due to disease or impairment. Because organ systems function as interconnected entities rather than independent units, their progression value can be considered as a latent continuous process from one system to another. Given this situation, we conceptualized BODN in the statistical models as a progressive ordinal metric in a Bayesian framework and postulated unequal intervals between its successive values. This approach provides an unequally spaced ordinal scale, where varying severities of diseases or disease stages can contribute flexibly to these unevenly distributed values. The predicted values of such models when focused on organ-specific diseases are called Bodily System-Specific Clocks (BSCs) and when combined to include all organ systems' disease levels produce a health entropy that we call the Body Clock. Therefore, the Body Clock is a more nuanced metric than BODN and is operationalized from each disease-weighted contribution into BODN for each person over time without including chronological age.

Bodily system health entropy can variably affect functional outcomes such as walking speed, as well as physical and cognitive disability. Therefore, the Body Clock subsequently informed additional tools to assess walking speed (the Speed-Body Clock) and a new Disability Index (Disability-Body Clock). To capture the rate of aging in terms of multidimensional health, we regressed chronological age on each set of clocks. Together, these eight metrics form the Health Octo Tool (Fig. 1), which provides multidimensional health assessment and aging rates which can operate independently of chronological age. By incorporating early disease states and assessing the impact of system entropy on functional outcomes, the Health Octo Tool offers a robust framework for understanding and tracking multidimensional pathologies and heterogeneity of aging across lifespan, supporting the evaluation of interventions aimed at improving healthy aging at both the population and individual levels.

## Results

Within each organ system, we defined disease states, and their severities based on accepted medically pre-defined disease criteria (Table S1), using combinations of medical history, medical laboratory and physical examinations. The related health questions are listed in the Table S2, and the disease definitions and their levels are also summarized in Table S1. The organ systems considered were cardiovascular (CV), renal (Re), metabolic (Me), gastrointestinal and liver (GL), respiratory (Res), Thyroid (Th), hematopoietic (He), oral health [i.e., periodontitis (Pe)], musculoskeletal (MS), sensory (Se), and central nervous system (CNS).

### Body organ disease number (BODN)

BODN was determined as the number of organs with at least one disease, serving as a new measure of system morbidity. Pathology at the specific organ level was established based on predefined disease/ impairment criteria, either deviating from normal or as used for disease diagnoses in clinical practice (Table S1). For instance, the cardiovascular system encompasses diseases such as hypertension [defined using systolic and diastolic blood pressure above the cut-points], ischemic heart disease [clinically defined as developing chest pain when exercising and other related symptoms identified on the Electrocardiogram (ECG)], peripheral artery disease (measured by ankle-brachial index), different arrhythmias (based on ECG), and congestive heart failure (based on signs and symptoms such as shortness of breath, low ejection fraction value [<40], and other medical signs and symptoms), each varying in severity. We considered an organ system to have morbidity if it exhibited at least one of the organ-specific diseases or deviation from normal health (subclinical states in some of diseases like sub-clinical hypothyroidism). It is important to note that disease diagnosis in the medical field relies on a combination of factors

including medical history, physical examination, organ-specific laboratory tests, and ongoing prescribed treatments. The questions and medical examinations are summarized in Tables S1 and S2. Therefore, BODN is derived by combining all this clinical information. The definition of BODN, reflecting progressive system morbidities, using the 11 organ systems was then extended to include cerebrovascular accidents (CA), and cancer (Can) (Fig. 1, S1, Tables S1, S2), because CA involves multiple mechanisms, such as hypertension, coagulopathy and genetics, and cancers have both shared and distinct mechanisms across organs. Therefore, we considered them as separate system entities from other organ systems.

Since organ systems operate as interconnected parts of a whole, their pathologies do not occur in isolation but progressively, with uneven transitions from one to the another, often across different systems. Considering these transitional states, BODN was considered as a progressive arithmetic measure, characterized by variable intervals between consecutive values. This design allows for an ordinal scale with cumulative family[28]. In Bayesian statistical models, cumulative family allows cut-points to be introduced to the continuous latent BODN using probability information from the data, where differing disease severities or stages contribute to these irregularly spaced cut-points, enabling quantification of variable contributions of disease severity or levels to its non-equidistant values. We employed the concepts of ordinal outcome with cumulative family in a Bayesian framework to model BODN[28]. Moreover, ordinal numbers were assigned to the disease levels as covariates of the models, representing the severity of each organ-specific pathology, serving as lagged predictors of BODN (e.g., no hypertension, hypertension with no treatment, and hypertension with treatment would be coded 1, 2, and 3, respectively; Table S1). A monotonic effect function described in Bayesian statistics was used to capture the maximum effect of disease as well as level-specific effects. This approach allows for detailed and nuanced quantification of disease effects on BODN[29].

## Health octo tool metrics

Through post hoc analyses using the posterior prediction function, we derived predicted values of BODN from the models that included organ-specific diseases as covariates and created Bodily System-Specific Clocks (BSCs). By considering all diseases collectively, we developed the Body Clock, a measure of health entropy that is obtained from prediction of the model, where all disease burdens were covariates to predict BODN over time without adding chronological age. Using Gamma distribution models with a log link function, we regressed chronological age against BSCs and the Body Clock to obtain Bodily System-Specific Age and Body Age, as proxies for organ-specific and whole-body systems rates of aging, respectively (Table S3). Walking speed is one of the phenotypes of aging most related to health in older persons. To understand the effect of whole bodily systems' health on walking speed, we estimated the effect of Body Clock on walking speed and the predicted value is called Speed-Body Clock. Capturing function and whole system aging heterogeneity, the chronological age was regressed over Speed-Body Clock to obtain Speed-Body Age.

We developed a Disability Index (DI) by applying the Zero Inflated Beta Binomial (ZIBB) approach in a probability theory framework[30] (Formula [S1] and [S2] in the Supplemental Methods), combining information on physical and cognitive function, falls, and urinary or bowel incontinence (Table S4). Regressing DI against the Body Clock allowed us to understand how body health entropy predicts DI and obtained the Disability-Body Clock from the predictive value of the model. Regressing chronological age over Disability-Body Clock using Gamma distribution family, its age estimator is Disability Body Age—a proxy for the rate of aging that incorporates bodily system health and disability states. Together, the four sets of clocks and their corresponding rates of aging comprise the Health Octo Tool (Fig.1). The equations and description of each model are summarized in Table S3.

We performed longitudinal data analyses in the Baltimore Longitudinal Study of Aging (BLSA)[31,32] and replicated the analyses using the Invecchiare in Chianti (InCHIANTI) study[33]. Cumulative multilevel ordinal regression was employed to estimate the posterior coefficient values for lagged diseases contributing to BODN[28,30,34,35]. To manifest increasing weight of health entropy compared to chronological age, various models were developed, including those for chronological age, single diseases, single system diseases, multiple-system diseases, and global-system entropy, including all systems as co-variates. Model performance was assessed using leave-one-out cross-validation (LOO-CV)[36], and the model weights were compared with the chronological age model as a reference using "stacking" in the Bayesian framework to evaluate model performance[37].

## Validation assessment using BLSA parameters as a training set applied to InCHIANTI and NHANES data as test sets

In the Bayesian framework, employing parameters from the training data (BLSA) and applying them to new data provides out-of-sample predictive accuracy. The parameters of the BLSA entropy model, fitted with BLSA data, were tested by making predictions for individuals in the InCHIANTI study($n = 986$, women = 551) and subsequently for 40,700 participants in the National Health and Nutrition Examination Survey (NHANES) data from 2003 to 2018[38]. We compared the observed values with the predicted Body Clock using the visualization function pp_check from the brms package. Additionally, we examined the relationship between the Body Clock derived from the replicated analysis of InCHIANTI and the validated version obtained by applying BLSA data parameters to the InCHIANTI data.

The characteristics of the BLSA study by age group are summarized in Table S5. The schematic representation of BODN systems and combinations of systems' morbidities are shown in online interactive graphs (Body Organ Disease Patterns (shinyapps.io), S1, and S2 (Supplemental Results). The graphs illustrate a wide heterogeneity in system combinations within BODN, which increases with age. This pattern aligns with health entropy, characterized by the random expansion of health deterioration.

To demonstrate that entropy increases with the addition of diseases and affected systems, we initiated the models with longitudinal BODN as the ordinal outcome and lagged single diseases as ordinal covariates (predictors/features). We then sequentially added diseases within a single organ system (organ-specific diseases are defined in Table S1), multi-system diseases, and, ultimately, whole entropy, incorporating all organ-system diseases as covariates.

Analyses of the BLSA data and then InCHIANTI as replication revealed significant and varied effects of single diseases and organ system-specific diseases on the accumulation of system multimorbidity, as expressed by the longitudinal BODN (Table S6). Comparing the effects of organ system-specific diseases to those of individual diseases in the regression models, we found that, in most cases, single diseases had a greater contribution to BODN than the aggregated organ systems' diseases. For example, in the BLSA study, peripheral artery disease (PAD) contributed to BODN with posterior estimate [b]=3.43, 95% CI: 2.05–4.8, while the cardiovascular (CV) system's contribution was $b = 1.21$, 95% CI: 0.03–2.46 (Table S6A–B). Similarly, in the InCHIANTI data, the effect size of PAD as a single disease was b = 1.56, 95% CI: 1.06–2.06 (Table S6G–H), which was attenuated within the organ system ($b = 0.8$, 95% CI: 0.8–1.26) (Table S6G–H), indicating the link between diseases within the same organ system and potential shared mechanisms. Separately, we analyzed the effect of chronological age on BODN, and it emerged as a robust predictor of BODN (BLSA, time-1: $b = 0.20 \pm 0.01$, 95% CI = 0.18–0.22; time-2: $b = 0.24 \pm 0.01$, 95% CI = 0.21–0.26; InCHIANTI: $b = 0.14$, 95% CI = 0.12–0.14). Comparing model performance weights, called average model weight stacking (Supplemental Methods), indicated that the model including the chronological age as only predictor,

**Single Disease, Single System, and Multiple Systems Weights Vs. Age Only Weight Predicting BODN**

**Fig. 2 | Model weight comparisons using average model stacking.** Model weights for single diseases in the BLSA at time 1 (**A**) and time 2 (**B**), single systems (**C**, **D**), and multisystem (**E**, **F**) are compared to the corresponding age-only model weights. The data illustrate that the weight of single systems surpasses that of single diseases. As the number of systems increases, indicating a rise in entropy, the weight of multisystem models significantly surpasses that of chronological age, enabling a more optimal model for the prediction of longitudinal BODN. Sample size *n* = 907, Women = 451. The detailed weights are reported in Table S7.

the age-only model, outperformed individual disease or single organ-system models (Fig. 2A–D; Table S7), highlighting the necessity of developing models that explain the variability of health entropy with aging beyond what is explained by a single disease. On the other hand, these findings underscore that chronological age still strongly affects the deterioration of whole body health in a way that cannot be simply captured by single pathological or physiological measures.

### Multiple systems with morbidity outperform chronological age in predicting longitudinal BODN

Stepwise analyses that included multiple-systems diseases with increasing entropy to predict longitudinal BODN compared to chronological age alone substantially and significantly maximized and explained higher levels of entropy. This was evident in the superior model performance (larger ELPD: expected log pointwise predictive density) and greater model weights compared to chronological age-only models (Fig. 2E, F, Table S7). The higher ELPD indicates better predictive performance, as the model is more accurate in predicting observations. Furthermore, the full models, including all systems, outperformed chronological age in predicting longitudinal BODN in both the BLSA and InCHIANTI datasets (Fig. 2E, F, Table 1, Table S7). This suggests that capturing the health state of all organ systems as a comprehensive measure, referred to as whole-body entropy, can serve as a proxy for intrinsic aging.

In the BLSA dataset, incorporating all body disease burdens at time-2 provided a better prediction of BODN compared to chronological age (Fig. 2F). The full entropy model at time-2 exhibited higher model weights and ELPD than the time-1 model, indicating that the predictive power of intrinsic biological age increases with advancing chronological age (Table 1).

**Table 1 | Model assessments, model comparisons, and model weights to predict BODN**

| Model Fits | ELPD | SE | PSIS κ < 0.7 | ELPD_DIFF | DIFF_SE | Weights % |
|---|---|---|---|---|---|---|
| BLSA | | | | | | |
| Time1[a] | −3926.5 | 40.5 | 100.0% | −168.3 | 24.2 | 0.0 |
| Time-1 age-only | −3891.0 | 40.0 | 98.0% | −133.2 | 25.9 | 25.0 |
| Time-1 full entropy | −3758.0 | 42.3 | 100.0% | 0.0 (Reference) | 0.0 (Reference) | 75.0 |
| Time2[b] | −2830.0 | 33.3 | 99.9% | −384.7 | 39.3 | 0.0 |
| Time-2 age-only | −2751.3 | 32.6 | 99.8% | −305.4 | 29.5 | 10.0 |
| Time-2 full entropy | −2445.9 | 38.1 | 100.0% | 0.0 (Reference) | 0.0 (Reference) | 90.0 |
| InCHIANTI | | | | | | |
| Time | −5363.2 | 39.7 | 100% | −463.6 | 34.5 | 0.0 |
| Time age-only | −5174.2 | 39.4 | 100% | −274.5 | 35.50 | 10.0 |
| Full entropy | −4899.6 | 46.4 | 100% | Ref | | 90.0 |

*ELPD* expected log posterior predictive density (the higher the value, the better model is), *ELPD-DIFF* difference in ELPD values compared the time, the age-only, and full entropy models, *SE-DIFF* standard error of the ELDP_DIFF.
[a]Time from baseline to the end of study.
[b]Time from the second visit to the end of study.

## Bodily System-Specific Clocks (BSC) and Bodily System-Specific Ages (BSA)

Given the diverse contribution of different organ systems to the BODN for each person, our study further explored heterogeneous organ-specific intrinsic aging as clocks, when organ-specific diseases are predictors of BODN, and rates of aging, when chronological age is regressed over such clocks. We propose the concept of Bodily System-Specific Clocks (BSC) and Bodily System-Specific Age (BSA) to underline the idea that human health is shaped by the entropic accumulation of damage in cellular structures and functions that act at the overall body organ system level (entropy that drives the rate of organ system aging, captured by BSA), as well as predisposition to specific organ pathology due to genetic susceptibility or specific environmental exposures (captured by BSC). Indeed, examination of correlations among the 13 organ-system BSAs revealed varying degrees of correlation between each pair of BSA values, indicating disparate aging rates among organ systems and suggesting that organs do not age synchronously (Fig. S3A, B).

The ordinal cumulative regression model was built using weak priors (in the Bayesian formula, the posterior coefficient is quantified via multiplication of prior knowledge on the coefficient by the likelihood of the coefficients, Supplemental Methods). The Bayesian posterior inference model was sampled using a Markov chain Monte Carlo approach (Supplemental Methods)[30]. In the BLSA study, BSA of several systems, including kidney, sensory, cardiovascular, musculoskeletal, metabolic, stroke, CNS, GI, liver, and thyroid, were consistently older than chronological age and even exceeded the reported maximum chronological age (Fig. 3). These results emphasize a biological aging process at the organ level that extends beyond 125 years, further suggesting that the emergence of pathology in human life results from the accelerated aging of specific organ systems. Interventions to delay aging may focus on reestablishing harmonious aging across different systems (Fig. 3). Notably, a similar trend emerged in the InCHIANTI dataset, confirming the replication of this approach (Fig. S3C).

To assess the degree to which each disease level contributes to BODN, we quantified posterior coefficient estimates of each disease level and 95% credible intervals using a monotonic effect in the Bayesian approach. The detailed monotonic effect function is described in the Supplemental Methods[29]. In both BLSA and InCHIANTI datasets, collectively including all diseases of organ systems led to a heterogeneous and significant contribution to BODN. However, for some organ-system diseases, such as peripheral artery disease, hyperthyroidism, thrombocytopenia, adult-onset asthma, and Parkinson's disease, there was substantial uncertainty in the estimate, reflected by wider 95% credible intervals, possibly because the prevalence of these

conditions in the population is relatively low (Fig. 4A, B, Fig. S4 A, B, Table S6C, S6F, I). Interestingly, milder states of organ system diseases (e.g., transitional ischemic attack, impaired glucose tolerance, mild anemia, subclinical hypothyroidism, cataract, mild liver disease, gingivitis without edentulous, osteopenia, mild osteoarthritis, and poor hearing that improves with a hearing aid) had larger estimates contributing to longitudinal BODN compared to more severe states of these diseases (Figs. 4A, B, S4 A&B, Table S6C, F, I). Similarly, diseases without pharmacological treatment (e.g., hypertension [HTN], type-2 diabetes mellitus, hyperlipidemia, chronic bronchitis, gastrointestinal disease, depression, and Parkinson's) had larger posterior coefficient estimates for predicting BODN (Fig. S4A, B Table S6). For example, HTN without treatment was a stronger contributor to BODN ($b = 0.32$, 95% CI: 0.23–0.39) than HTN with treatment ($b = 0.17$, 95% CI:0.13–0.22). Similarly, congestive heart failure with preserved ejection fraction, a common type of heart failure in older adults, had a larger posterior coefficient estimate ($b = 0.22$, 95% CI:0.12–0.23) compared to the congestive heart failure with a low ejection fraction ($b = 0.09$, 95% CI: 0.05–0.13). Arrhythmias, such as sinus bradycardia, elongated QTc, and atrial fibrillation, also significantly affected BODN (Fig. S4A, B; Table S6C, F, I). Stage-1 age-related chronic kidney disease (CKD), decoupled from diabetic kidney failure, exhibited stronger incorporation into BODN ($b = 0.53$, 95% CI: 0.38–0.71) compared to stage-2 ($b = 0.07$, 95% CI:0.05–0.10) and stage-3 CKD ($b = 0.1$,95% CI:0.05–0.10), as well as end-stage renal disease ($b = 0.20$, 95% CI:0.15–0.28). Similar results were observed in the InCHIANTI study for stage-1 CKD (Table S6). These patterns were also observed in the time-2 full model BLSA, where hyperthyroidism and Parkinson's significantly incorporated into BODN (Fig. S4A, B, Table S6). Overall, as pointed out above, model assessments using LOO-CV approach indicated that full entropy models, incorporating all disease severities, with larger ELPD (Table 1) had the best performance in predicting longitudinal BODN (Fig. S5). These findings suggest that the overall health assessment of an individual should consider pathology across a wide range of severities in all organ systems.

Interestingly, there was a slight discrepancy between the BLSA and InCHIANTI results. In the BLSA, peripheral artery disease (PAD) showed significant incorporation into BODN either as a single disease or as part of a single-organ system, but its significance was not retained in the full model; while in the InCHIANTI study, PAD retained its significance in the full model, albeit with a smaller magnitude (Fig. S4A, B, Table S6C, F, I). These findings suggest a potential shared pathophysiology among diseases within the cardiovascular system. This conclusion was further supported by comparing model weights using Bayesian Stacking[37] (Supplemental Method), which revealed that

**Fig. 3 | Relationships between bodily system-specific age (BSA) and chronological age.** The graph reveals that BSA for cardiovascular (CV), renal (Re), Res, CNS, and MS systems can exceed 120, surpassing the maximum reported human chronological age. System abbreviations same as Fig. 1. Sample size $n$ = 907, Women = 451.

model weights of 49.7%, 48.3%, and 2% for HTN, congestive heart failure (CHF), and arrhythmia, respectively, while the weights for PAD and ischemic heart disease (IHD) models were zero, indicating a higher degree of shared pathophysiology among certain cardiovascular-related diseases. In the InCHIANTI study, the weights for HTN, CHF, and arrhythmia were 58%, 41%, and 1%, respectively.

**Personalized Body Clock and Body Age**

We developed the individualized Body Clock by extracting the predicted value of BODN from the post hoc analysis of the multilevel ordinal regression model that included all organ systems diseases and their severities for each individual. Briefly, the Eqs. (1) and (2) in the brms R package are as follows, where mo stands for the monotonic function for the ordinal predictor:

$$
\begin{aligned}
\text{Fit\_BLSA} = (\text{bodn} \sim{}& \text{mo(Hypertension)} + \text{mo(congestiveHeartFailure)} \\
&+ \text{mo(IschemicHeartDisease)} + \text{mo(Arrhythmia)} + \text{mo(Kidney)} \\
&+ \text{mo(Diabetes)} + \text{mo(Hyperlipidemia)} + \text{PrepheralArteryDisease} \\
&+ \text{mo(Stroke)} + \text{mo(Anemia)} + \text{Thrombocytopnia} \\
&+ \text{mo(GastrointestinalDisease)} + \text{mo(Liver)} + \text{mo(COPD)} + \text{Asthma} \\
&+ \text{mo(OralHealth)} + \text{mo(Hypothyroisism)} + \text{Hyperthyroisism} \\
&+ \text{mo(OsteoArtheristis)} + \text{mo(Osteoporosis)} + \text{mo(Hearing)} + \text{mo(Eye)} \\
&+ \text{mo(Depression)} + \text{mo(sParkinsons)} + \text{mo(Cognition)} + \text{Cancer} \\
&+ \text{yrs} + (1 + 1|\text{id}), \text{data} = \text{BLSA})
\end{aligned}
$$

(1)

$$\text{Body Clock\_BLSA} = \text{posterior\_predict(fit\_BLSA, data} = \text{BLSA})$$

The above equation was applied to the InCHIANTI data for replication, and the predicted model Eq. (2) is:

$$\text{Body Clock\_InCHIANTI} = \text{posterior\_predict(fit\_InCHIANTI, data} = \text{InCHIANTI})$$

(2)

Overall, the median Body Clock in the BLSA dataset was 6.1 (range: 2.1–11.5) and for InCHIANTI was 6.6 (range: 2.4–11.1). It is worth noting that the BLSA enrolls individuals who are very healthy at baseline, whereas the INCHIANTI dataset is population-based and primarily includes individuals over the age of 50. In the InCHIANTI study, women 65 to 85 years old exhibited a higher Body Clock value than men, with measurements of $7.1 \pm 1.1$ compared to $6.8 \pm 1.2$, respectively. However, in the BLSA study, there were no sex differences in Body Clock values.

We used the monotonic effect to assess severity-specific coefficients and derive the latent maximum disease severity effects, which form a linear scale from minimum to maximum effect, to obtain the Body Clock at the individual level (Supplemental Methods). Furthermore, individualized trajectory plots for Body Clock vividly demonstrate diverse trajectories among individuals of the same chronological age. These trajectories exhibit varying slopes and magnitudes, underscoring the unique maximum intrinsic aging experienced by different individuals (Fig. S6A, B). Additionally, age was categorized into the following groups: <45, 45–54, 55–64, 65–74, 75–84, and 85 and older. The Body Clock increases with each successive 10-year age interval,

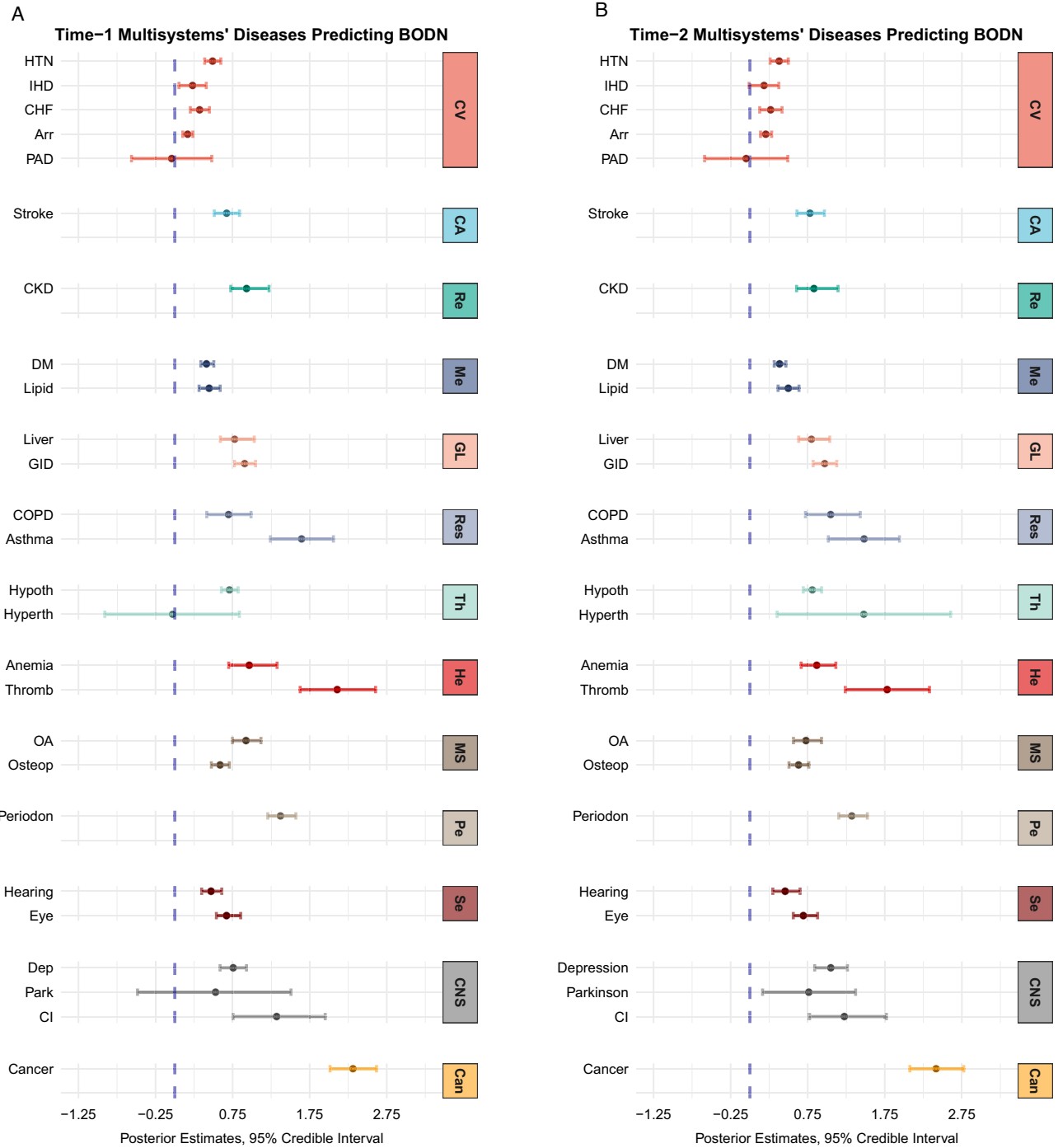

**Fig. 4 | Lagged full-model multisystem posterior estimates predict longitudinal body organ disease number (BODN).** The lagged full-model estimates reveal heterogenous contributions of diseases into BODN. **A** Maximum contribution of diseases into longitudinal BODN at time 1. **B** Maximum contributions of diseases into longitudinal BODN at time 2. Larger posterior estimates, accompanied by narrower 95% credible intervals (CI), yield more accurate predictions. The maximum effects of most diseases increase with time. Significance is determined when the 95% CI does not include 0. Disease Abbreviations: HTN hypertension, IHD ischemic heart disease, CHF congestive heart failure, Arr arrhythmia, PAD peripheral artery disease, CKD chronic kidney disease, DM diabetes mellitus, GID gastrointestinal disease, COPD chronic obstructive pulmonary disease, Hypoth hypothyroidism, Hyperth hyperthyroidism, OA osteoarthritis, Osteop osteoporosis, Periodon periodontal disease, Dep depression, Park Parkinson's disease, CI cognitive impairment. System abbreviations: same as for Fig. 1. For the values of mean posterior estimates and 95% CI, please see the values in Table S6C, F for full models. Sample size $n = 907$, Women = 451.

and its values are heterogeneously distributed within each age group, as depicted in both the BLSA and InCHIANTI studies (Fig. S7A, B). These findings emphasize the diversity of the aging process. Subsequently, the chronological age was regressed over the Body Clock using a Gamma distribution family and a log link function, allowing the fitting of non-linear, exponential points into the models (equations [3] and [4]).

$$\text{Fit}_{\text{Age}} = \text{Chronological Age} \sim \text{Body Clock}, \text{data} = \text{BLSA}, \quad (3)$$

$$\text{Body Age} = \text{posterio predict}(\text{Fit}_{\text{Age}}) \quad (4)$$

The same formula was applied to the InCHIANTI data.

We obtained the post hoc prediction for each individual using posterior predict, termed Body Age (Eq. 4, Fig. S7C, D). At the population level, given the constant value of 3.78 and the mean posterior estimate of 0.12 obtained from the Bayesian Gamma distribution model with BLSA, where chronological age was regressed over Body Clock, if an individual has a Body Clock value of 9, the Body Age would be 129 (calculated as e$^{(3.78+0.12\times9)}$ = 129.02, e is the natural logarithm). However, it is crucial to avoid generalizing the population mean posterior estimate, rather Body Age should be quantified for an individual, obtained from the prediction model using individual-specific posterior estimates, to ensure accurate predictions for precision-medicine decisions.

## Speed-Body Clock and Speed-Body Age
Walking speed serves as a fundamental measure of health commonly utilized as an important risk factor, biomarker, and outcome at the old age. Walking speed was recorded (in meters per second) over 6 meters in the BLSA and 7 meters in the InCHIANTI studies. To assess the influence of the Body Clock on walking speed, we predicted longitudinal changes of walking speed by the Body Clock, adjusting for height using a Gaussian regression and modeled the walking speed variance (sigma) over the Body Clock to develop another health metric that we called Speed-Body Clock (equation [5] and [6]; Fig. S8 A, B).

$$Fit\_BLSA = Walking\ Speed \sim Body\ Clock + height, data = BLSA \quad (5)$$

$$Speed - Body\ Clock = Postrior\ predict(Fit\_BLSA) \quad (6)$$

The same formula was applied to the InCHIANTI study for replication.

Because of the inverse relationship between the Body Clock and walking speed, a smaller Speed-Body Clock indicates a negative impact of system's health on physical function. In the BLSA study, the Speed-Body Clock was lower in women compared to men, even after adjustment for height, with values of $1.09 \pm 0.1$ for women and $1.15 \pm 0.1$ for men. At the population level, for each unit increase in the Body Clock, the posterior coefficient of walking speed decreased by 0.062 m/s (0.065–0.062) in men and 0.064 m/s (0.063–0.061) in women (Fig. S8A). In the InCHIANTI study, a one-unit increase in Body Clock corresponded to a decrease in the posterior coefficient of walking speed by -0.03 m/s in both sexes. Again, women exhibited a smaller Speed-Body Clock than men, with values of $1.29 \pm 0.15$ for women and $1.56 \pm 0.14$ for men (Fig. S8B). For BLSA, the mean Speed-Body Clock was significantly different between age groups so that the youngest group (<45 years old) with a 1.26 value had the largest, and the oldest group with 1.01 value had the smallest Speed-Body Clock. Similar results were observed in the InCHIANTI study with 1.55 vs. 1.29 values for the youngest and oldest groups, respectively. These findings indicate that bodily system health entropy (Body Clock) can contribute to a decline in walking speed, which accelerates after age 65.

Additionally, individuals of the same chronological age may have different rates of aging in terms of how the Body Clock affects walking speed. This heterogeneity in the aging rate is captured by regressing chronological age on the Speed-Body Clock using a Gamma distribution and the predicted value was denoted as Speed-Body Age (equation [7] and [8]; Fig. S8 C, D).

$$Fit\_BLSA = Age \sim Spreed - Body\ Clock + Height, data = BLSA \quad (7)$$

$$Speed - Body\ Age = psoterior\ predict(Fit\_BLSA) \quad (8)$$

The same formula was applied to the InCHIANTI data for replication.

With a one unit increase in the Speed-Body Clock, the Speed-Body Age decreases in both BLSA (intercept=5.34, $b = -0.99$, 95% CI (−1.06 to −0.95)) and InCHIANTI (intercept = 4.8, $b = -0.43$, 95% CI = −0.48 to −0.39). That is, using a Gamma distribution interpretation, we predicted that a Speed-Body Clock of 0.8 resulted in a speed Speed-Body Age of e$^{(4.8-0.43*0.8)}$ = 86.14 in InCHIANTI and 94.44 in BLSA. The difference between these estimates may be explained by the healthier people enrolled in BLSA, while InCHIANTI is a population-based sample. A larger Speed-Body Age represents resilience to functional or system function decline and can reflect aging heterogeneity. Note that we relied on individual-based metrics using a Bayesian approach that can be obtained from prediction models rather than the population level metrics and thus reduced the chance of bias due to population variability. In the InCHIANTI study, the Speed-Body Age could extend up to 105 units, while in the BLSA, the maximum Speed-Body Age did not surpass 97 units. Despite the smaller average population state within InCHIANTI, some individuals manifested resilience indicated by a higher Speed-Body Age. The data revealed increasing heterogeneity in Speed-Body Age after age 70 as depicted in Fig. S8C, D.

## Disability index (DI), Disability-Body Clock, and Disability-Body Age
To capture late-onset outcomes experienced by individuals, we devised a new Bayesian DI, in which each individual could have multiple components of physical and cognitive disability. We incorporated a total of 47 measured components as the total number of trials in the DI (Table S4). The components representing these outcomes were modeled using the Zero Inflated Beta Binomial (ZIBB) distribution, estimating the probability distribution of the number of observed components (events) given the total number of measured components (trials) for each individual. This approach allowed for flexible probabilities that could vary from person to person. Moreover, ZIBB handled model instability resulting from zero values in young populations without age-related disability or in the older adults exhibiting resilience to late-onset outcomes. The detailed ZIBB formula is described in the Supplemental Methods (equations [S1], [S2], and below, equations [9] and [10]).

$$Fit\_BLSA = probability\ of(events)|trial(total\ events) \sim 1 + (1|id), data = BLSA$$
$$(9)$$

$$Disability\ Index = posterior\ predict(Fit\_BLSA) \quad (10)$$

The same formula was applied to the InCHIANTI data for replication. This approach enabled us to explore how the Body Clock predicts the model-based physical and cognitive DI, based on events within the total 47 measured (trials) components (Table S3).

Not everyone with an elevated Body Clock experiences disability. This heterogeneous process can be captured using the Disability-Body Clock (defined as the sum of positive events conditioned on the total number of measured events and predicted by the Body Clock). This approach enabled us to explore how the Body Clock predicts the model-based physical and cognitive DI, based on events within the total 47 measured components (Table S4; equations [11 and [12]).

$$Fit_{BLSA} = probability\ of\ ((events))|trial(total\ events) \sim Body\ Clock + (1|id), data = BLSA$$
$$(11)$$

$$Disability - Body\ Clock = posterior\ predict(Fit\_BLSA) \quad (12)$$

The same formula was applied to the InCHIANTI data for replication.

In the BLSA data, the median DI was 0.03 (range 0.007–0.71). The median Disability-Body Clock was 1.2 (range 0.12–9.48) and not different between men and women (women: 1.5 ± 1.04; men: 1.52 ± 1.08) (Fig. S9.A). In the InCHIANTI study, the median DI was 0.22 (range 0.05–0.95) with a larger mean DI in women (0.3 ± 0.16) than men (0.22 ± 0.13). The median Disability-Body Clock was 6.3 (range: 1.45–12.8) with a larger mean in women (6.5 ± 1.82) than men (6.17 ± 1.74) (Fig. S9B). Additionally, individuals of varying ages may exhibit different effects of the Body Clock on disability. To account for this heterogeneity in such aging rate, we regressed the chronological age against the Disability-Body Clock. The predicted value from this model is referred to as the Disability-Body Age (equations [13] and [14]).

$$Fit_{BLSA} = Age \sim Disability - Body\ Clock, data = BLSA \qquad (13)$$

$$Disability - Body\ Age \sim posterior\ predict(Fit\_BLSA) \qquad (14)$$

The same formula was applied to the InCHIANTI data for replication.

In the BLSA data, the median Disability-Body Age was 62.71 (range 60.4–135.67) with high heterogeneity after age 60. Moreover, in some individuals of either sex, the biological age surpassed 120 (Fig. 5). In the InCHIANTI data, the median Disability-Body Age was 67.3 (range 43.5–120.2) with women exhibiting a higher average biological age compared to men (69.56 ± 11.55 vs. 67.2 ± 10.7) (Fig. S9C).

## Body Clock predicts binary short physical performance battery (SPPB), disability, geriatric syndrome, and mortality

To evaluate the predictive capability of the Body Clock for age-related late-life binary outcomes, we utilized Multilevel Negative Binomial Regression models within the Bayesian framework, accounting for time. We focused on outcomes including SPPB < 9, disability (Activities of Daily Living scale [ADL] >1 and/or Instrumental Activities of Daily Living scale [IADL] >1), geriatric syndrome (injurious fall or urinary/bowel incontinence as dichotomous variables), and mortality. Our analysis revealed that individuals with higher Body Clock scores had an elevated risk of SPPB < 9 (BLSA: hazard ratio [HR] = 2.7, 95% CI = 1.7–10.6; InCHIANTI: HR = 2.0, 95% CI = 1.7–2.3), geriatric syndrome (BLSA: HR = 1.6, 95% CI = 1.5–1.7; InCHIANTI: HR = 1.2, 95% CI = 1.2–1.3), disability (BLSA: HR = 1.7, 95% CI = 1.5–1.8; InCHIANTI: HR = 1.6, 95% CI = 1.5–1.7), and mortality (BLSA: HR = 1.8, 95% CI = 1.5–2.2; InCHIANTI: HR = 1.5, 95% CI: 1.4–1.6). These findings indicate that the Body Clock could significantly predict age-related late-life outcomes.

The Body Clock superseded Frailty Index to predict late-onset binary outcomes. We used the same list of diseases described in Table S1 to manually develop disease-based FI scores, which is a common metric used to capture age-related deficits[11,39,40]. Rather than considering different levels of disease severity, each condition was treated as a separate binary variable. For example, in the case of ischemic heart disease, angina pectoris and acute myocardial infarction are considered distinct deficits in the FI, whereas the Body Clock treats them as different levels of the same condition. Similarly, for eye diseases, the Body Clock treats cataract, glaucoma, and macular degeneration as varying levels of eye disease, while in the FI, each condition is treated as a separate binary variable (e.g., having cataracts vs. no cataract). Overall, the same diseases were considered in both the FI and Body Clock. The primary difference lies in the methods (Bayesian-based Body Clock vs. manually developed FI) used to develop these health metrics.

It is important to note that while the FI and Body Clock exhibit strong correlation ($r = 0.83$, 95% CI: 0.816–0.84), the Body Clock more effectively captures health heterogeneity, as evidenced by instances where identical FI values correspond to varying Body Clock values (Figs. 6, S10A, B). We utilized Receiver Operating Characteristic (ROC) and Area Under the Curve (AUC) analyses to compare the predictive performance of the FI and Body Clock for various binary outcomes including SPPB, geriatric syndrome, disability, and mortality. The AUC values consistently indicate that the Body Clock provided stronger predictions for all these outcomes across both the BLSA and InCHIANTI studies (Figs. 7, S11A).

We used a Bayesian model for the FI and Body Clock to predict Bayesian DI. To assess DI predictions, we compared the Body Clock's performance comparing ELPD, which averages the logarithm of predictive densities for each data point and serves as a metric for comparing models, indicating how effectively they predict observed or new data points. Higher ELPD values signify better predictive performance and discern which model offers the most accurate predictions for the data. Our results demonstrated that the Body Clock exhibited better model performance compared to FI, with differences in ELPD favoring the Body Clock, indicated by the larger ELPD with both BLSA and InCHIANTI data. The ELPD difference presented as negative values in differences of ELPD (ELPD_DIFF) from the FI and Body Clock models where the Body Clock with the higher ELPD is the reference model (FI in BLSA: ELPD_DIFF = −27.9 ± 7.5; FI in InCHIANTI: ELPD_DIFF = −26.6 ± 6).

To gain deeper insights into the relationship between the Body Clock and SPPB, we explored the association between SPPB < 9 at time-1 and the Body Clock. This analysis revealed a reciprocal relationship between lagged functional impairment. With BLSA, we observed that SPPB < 9 was associated with a higher Body Clock value ($b = 1.9$, 95% CI = 1.6–2.4). Similarly, in the InCHIANTI dataset, SPPB < 9 predicted an increased Body Clock value ($b = 2.1$, 95% CI = 1.6–2.8). These results underscore the reciprocal nature of the association between the Body Clock and SPPB, suggesting when impaired physical performance is established it can significantly contribute to health deterioration as reflected in the Body Clock.

## Replication of Body Clock using NHANES data

We applied the same statistical pipeline described for BLSA and InCHIANTI to predict BODN in 40,700 individuals in the NHANES data, spanning the years 2003–2018, and derived personalized Body Clocks. As chronological age increased, the Body Clock demonstrated heightened heterogeneity after age 60, mirroring the pattern observed in the BLSA and InCHIANTI studies (Fig. S12A). The distribution of the BODN number suggests that morbidity in more than 5 organs becomes dominant after age 50 and older. However, some older adults with low Body Clock values exhibited bodily system resilience. In the NHANES data, women exhibited larger Body Clock values than men, suggesting that in general, women experienced poorer health compared to men. In addition, in NHANES data, Body Clock outperformed FIs specifically in predicting mortality, with an accuracy of 80% (Fig. S13).

The chronological age was then regressed over the Body Clock. Similar to the BLSA and InCHIANTI data, the Disability-Body Age surpassed the reported ceiling in humans (Fig. S12B). The Body Clock predicted SPPB, disability, geriatric syndrome, and all-cause mortality (Table 2). Of note, in the NHANES data, there was only a small sample size for adults older than 85 years and, thus, it is likely that our estimates were affected by information censoring.

## Applying the BLSA parameters (training set) to the NHANES and InCHIANTI data (validation sets) to develop the Body Clock

Using out-of-data input, the validation is the same concept as applying training data parameters (BLSA model) to a new data (InCHIANTI or NHANES) as test sets to obtain the Body Clock. We used the BLSA full entropy model to predict out-of-data input in the NHANES and the

### Disability-Body Age and Chronological Age in BLSA Data

Men

Women

**Fig. 5 | Disability-body age and chronological age in BLSA datasets.** Disability-Body Age surpasses the reported chronological age in humans. There is increasing heterogeneity in Disability-Body Age with increases in chronological age. Sample size $n$ = 907, Women=451.

### Frailty Index and Body Clock in BLSA Data

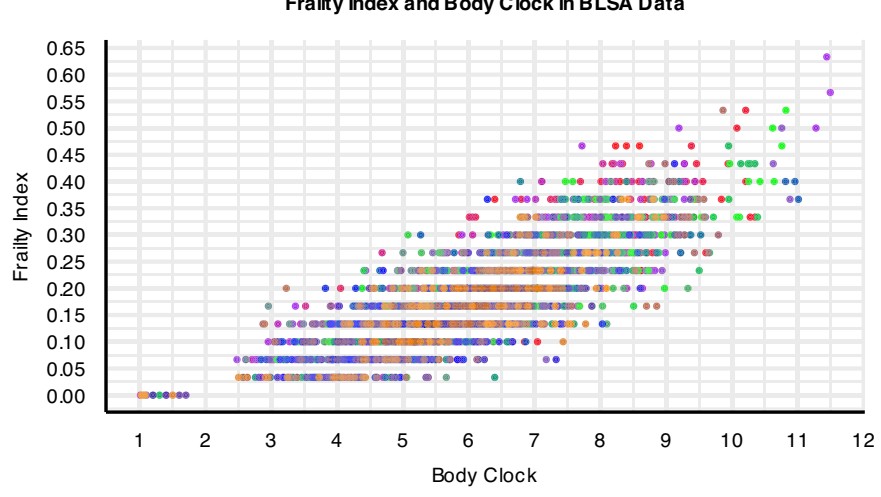

**Fig. 6 | High correlation between the Body Clock and disease-based frailty index (FI) with BLSA ($r$ = 0.83, 95% CI: 0.816−0.84).** Body Clock also captures heterogeneity better than FI: for heterogeneous values of the Body Clock−serving as a proxy for the intrinsic age and health entropy−FI values remain largely unchanged. Sample size $n$ = 907, Women = 451.

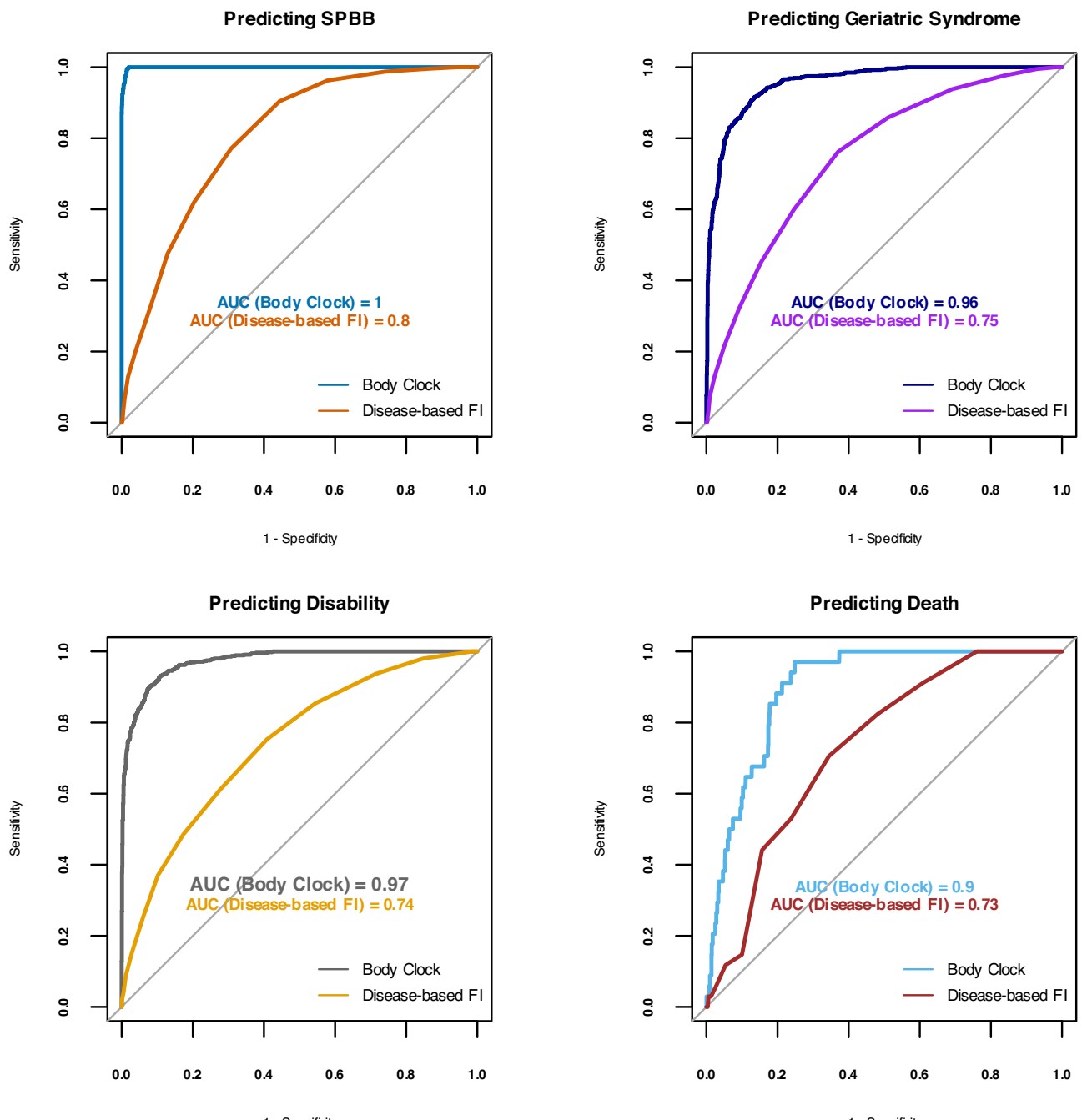

**Fig. 7 | Receiver operating characteristic (ROC) and area under the curve (AUC) of the Body Clock and FI score predict binary outcomes in the BLSA data.** The Body Clock predicts binary SPPB < 9, geriatric syndrome, and disability, superseding the FI score with more than 90% accuracy. Sample size $n = 907$, Women=451.

InCHIANTI data (test sets). The predicted and observed values in both studies were matched as depicted using posterior predicted check in the Bayesian framework (Fig. S14 A, B).

$$Fit_{BLSA} = BODN \sim mo(Disease1)\cdots + mo(DiseasesN) + time + (1|id), data = BLSA \tag{15}$$

$$\text{Predict in InCHIANTI(test set)} = \text{posterior predict(Fit\_BLSA, data = INCHIANTI)} \tag{16}$$

$$\text{Predict in NHANES(test set)} = \text{posterior predict(Fit\_BLSA, data = NHANES)} \tag{17}$$

There is a strong correlation between the Body Clock derived directly from the analysis of the InCHIANTI dataset (as a replication) and the Body Clock obtained by applying the BLSA parameters to the InCHIANTI dataset (as a validation test set) (Fig. 8). These results suggest that a valid model can be used to estimate the Body Clock in new data as a test set.

## Discussion

Using Bayesian inference on longitudinal data containing well-defined comprehensive clinical information, we developed and validated a personalized health tool that aligns multidimensional health with accelerated aging. Considering multimorbidity as a comprehensive measure of organ system health, we introduced the concept of BODN.

**Table 2 | Body Clock predicts binary health outcomes in NHANES data over 16 years**

| NHANES | SPPB category HR (95% CI) | Urinary Incontinence | Disability | Mortality |
|---|---|---|---|---|
| 2003–2004 | 1.67 (1.60–1.70) | 1.05 (1.02–1.08) | 1.90 (1.84–1.95) | 1.60 (1.45–1.77) |
| 2005–2006 | 1.71 (1.65–1.79) | 1.09 (1.06–1.14) | 1.86 (1.79–1.93) | 1.57 (1.40–1.77) |
| 2007–2008 | 1.67 (1.60–1.72) | 1.12 (1.09–1.15) | 1.68 (1.65–1.72) | 1.55 (1.45–1.68) |
| 2009–2010 | 1.68 (1.63–1.75) | 1.12 (1.08–1.14) | 1.90 (1.84–1.93) | 1.67 (1.52–1.84) |
| 2011–2012 | 1.77 (1.70–1.86) | 1.09 (1.06–1.12) | 1.78 (1.72–1.82) | 1.61 (1.48–1.75) |
| 2013–2014 | 1.77 (1.70–1.82) | 1.17 (1.15–1.21) | 1.95 (1.90–2.01) | 1.70 (1.52–1.77) |
| 2015–2016 | 1.82 (1.77–1.90) | 1.14 (1.1–1.17) | 1.97 (1.92–2.03) | 1.62 (1.49–1.77) |
| 2017–2018 | 1.71 (1.66–1.77) | 1.16 (1.14–1.20) | 1.73 (1.68–1.79) | 1.58 (1.45–1.75) |

*HR* Hazard ratio, *CI* credible interval.

The random and wide combinations of systems in the BODN represent the random expansion of health deterioration representing health entropy. We quantified the contribution of each disease severity over time using Bayesian models. Notably, we showed sub-clinical pathologies, such as sub-clinical hypothyroidism or mild kidney function decline, can significantly contribute to health entropy. One interpretation is that mild disease is part of the aging process while greater severity could be related to a specific organ system's susceptibility, from genetics as well as behavioral and environmental factors, that accelerate consumption of allostatic responses. We developed system-based clocks (BSCs) and their corresponding rates of aging (BSAs), demonstrating that organs age heterogeneously and can exhibit different biological ages, even at the same chronological age.

There are some population-based discrepancies in biological organ age, underscoring the diverse nature of the aging process across organ systems in different environments, which also can be potentially influenced by different genetics and behavioral factors[41]. This finding emphasizes the necessity of using individualized health metrics.

Including all disease burdens that contribute to BODN, we developed the Body Clock as a weighted BODN. This entropy model exhibits better prediction of BODN compared to age-only, single disease, or single-system models manifesting increasing health entropy burden. While chronological age remained a significant predictor of BODN, integrated disease levels of body organ systems surpass chronological age in predicting future BODN, revealing multimorbidity as a clinical manifestation of the intrinsic rate of aging that would otherwise remain concealed when focusing solely on individual diseases or organ systems. These findings strongly support the idea that as we age, chronological age becomes a weaker predictor of global bodily health. Instead, the emergence of pathologies, particularly multiple sub-clinical or mild impairments, becomes a more effective metric for measuring the rate of aging. In this framework, we were able to predict individual-based BODN and quantify the Body Clock as a measure of intrinsic biological aging independent of chronological age. The Body Clock serves as a strong predictor of late binary outcomes and supersedes the FI for predicting disability, geriatric syndrome, functional decline, and mortality. Moreover, as an individual-based measure of damage accumulation, the Body Clock enables the differentiation of biomarkers between those indicating accumulation of damage and those indicating resilience at the organ and whole-body levels. Regressing chronological age over the Body Clock, we determined the rate of aging in the whole-body systems; in another word, we obtained the rate of aging in terms of health entropy, which we call Body Age. We used the Body Clock to predict walking speed, a powerful functional phenotype in older adults. The Speed-Body Clock represents the effect of the whole bodily systems health entropy on functional health, while Speed-Body Age quantifies the rate of aging in terms of the impact of the whole-body systems on functional health. Speed-Body Clock can potentially differentiate individuals with

resilience to functional decline, despite increases in the Body Clock, from those with poor function. Such a metric can be used to study resilience or assess responses to multidimensional health interventions. We also devised a new approach to disability called the Bayesian DI that captures both cognitive and physical disability using a Bayesian model that accounts for model uncertainty due to the number of zero values. Similarly, we used the Body Clock to predict DI, deriving predicted values as the Disability-Body Clock and its rate of aging, termed the Disability-Body Age, to reflect the impact of the whole-body systems' health entropy on combined functional and cognitive disability. Such metrics allow us to understand why some individuals age with better health and free of disability while some present accelerated disability with or without increases in the Body Clock.

Of the known multimorbidity indices, the Charlson Index[42] and the Cumulative Illness Rating Scale for Geriatrics (CIRS-G) are widely used as hospital-based multimorbidity indices[43]. The Charlson Index incorporates disease weights based on 1-year mortality hazard ratios and age categories, but it may introduce bias due to selective mortality and lack of consideration for early-onset changes in health. In contrast, the Body Clock is decoupled from chronological age and reflects progressive health impairments at any age, independent of mortality. The CIRS-G assesses acute and chronic disease burdens and disability but may skew severity scores towards acute conditions or late-onset outcomes, potentially underestimating chronic diseases at younger ages. As we show, the Body Clock can be employed to predict late-life outcomes at any age, offering potential benefits in clinical settings and for preventive strategies.

The FI is also commonly used to measure deficits and aging in geriatric research[11,44] and has repeatedly been shown to predict mortality and a number of other health outcomes[17,24,25]. Notably, increases in the prevalence of frailty with decreases or no change in mortality have been shown, suggesting the necessity of understanding and preventing frailty[45]. We believe that there are conceptual similarities in our approach and the frailty paradigm, mainly that aging implies a progressive accumulation of entropic damage, with the Body Clock capturing heterogeneous health states and more strongly predicting late-onset outcomes. Some FIs include components of late-onset outcomes and also assign equal weight to all deficits, disregarding their varying magnitudes of effects on the body. Including late-onset deficits as components of disability tends to bias FI towards older age groups, resulting in an underestimation of early-onset aging. Therefore, in our study, to compare the FI with the Body Clock we consistently used the same diseases in both algorithms. Moreover, FI may not adequately consider the heterogeneity of disease impacts on the body and suffers from model instability and variability in the compositions of the components, affecting its reliability[46]. We demonstrated a strong correlation between the Body Clock and FI; nevertheless, FI lacks sensitivity to heterogeneity in the aging process. The methodology used to develop the Body Clock incorporates prior knowledge

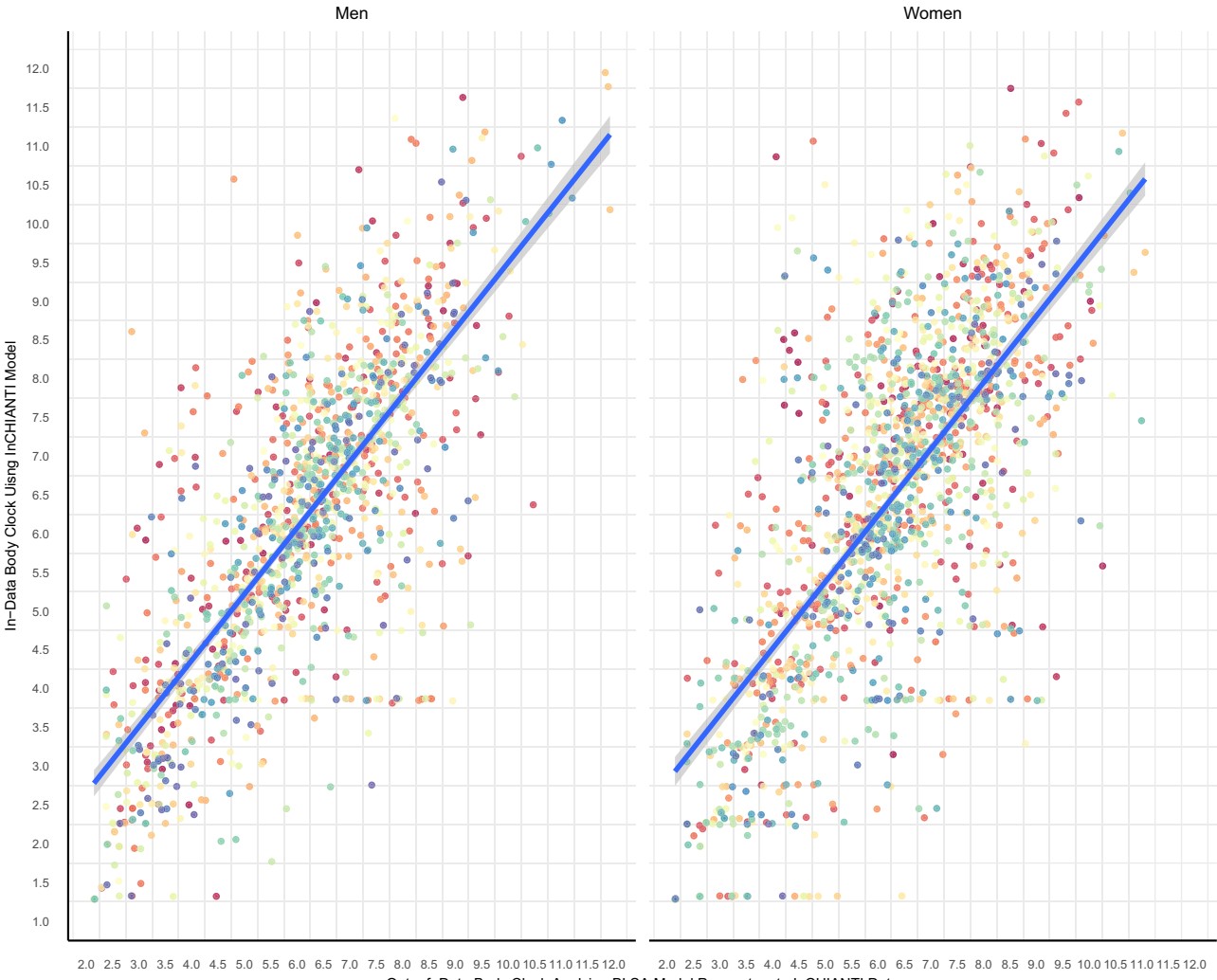

**In−Data (Replication) InCHIANTI Body Clock vs. Out−Of−Data (Test Set) Body Clock**

**Fig. 8 | There is a high correlation between in-data and out-of-data methods used developing the Body Clock in InCHIANTI study.** With the in-data method the full model (all diseases predicting longitudinal BODN) was performed, and the Body Clock (replication) was obtained. Using the out-of-data method the parameters of the BLSA model (training set) were applied to the InCHIANTI data (test set). The graph depicts a high correlation between in-data and out-of-data methods of developing the Body Clock. The Pearson correlation for men was $r = 0.76$, 95% CI (0.73–0.78) and for women was $r = 0.70$, 95% CI (0.68–0.72). The parameters of a valid model can be used to predict new data. Sample size in InCHIANTI data $n = 986$, women = 551.

and provides a robust framework for Bayesian analysis, avoiding the pitfalls of manual calculations.

Physical function is a phenotype of aging[47,48]. Some multimorbidity tools, like the multimorbidity-weighted index, use physical function to weight diseases[19,20]. However, this may introduce bias as compensatory strategies in some organs can offset damaging effects[4], just as resilience to functional decline undermines a disease's effect on other body systems. The Body Clock as an overall whole-body system health entropy measure predicts functional decline. We found that decline in physical function measured by SPPB also increases the value of the Body Clock, showing the bidirectional link between physical function and multimorbidity[49].

Certain indices use diseases to predict in-hospital mortality for multimorbidity. Mortality-based tools[21] may skew results towards fatal diseases. We quantified the Body Clock independent of mortality or other late-onset outcomes serving as an entropy of whole-body systems, while it predicts these outcomes. This finding suggests that capturing the health state of all bodily organ systems as a comprehensive measure (whole-body entropy) reflects intrinsic biological aging and predicts lifespan. Moreover, the components of the Health

Octo Tool emphasize that maximizing healthy longevity in humans might require a combination of interventions that target multiple aspects of health based on the basic mechanisms that prevent and repair damage accumulation with aging, as well as early diagnosis of particular pathologies driven by specific genetic and environmental factors.

Our data indicate that diseases at subclinical states or without treatment have significant predictive power for BODN. Clinical trials have linked even mild hypertension to cognitive impairments[50], and early intensive treatment may protect cognitive function and kidney health[51–53]. Our research supports the idea that sustained exposure to subclinical disease levels leads to chronic wear and tear, resulting in damage accumulation and variable organ impairment, as reflected in changes in the personalized Body Clocks. The large estimates and uncertainty associated with some disease states suggest that certain organs respond more readily to accumulated damage, possibly with reduced resilience. Therefore, the Body Clock, as an integrated estimate of disease levels contributing to BODN, can capture allostatic overload, representing cumulative physiological dysregulation beyond the body's ability to adapt to stress[4]. Thus, various body

stressors, such as diseases, medications, infections, and socio-environmental factors, may manifest as unique rates of increase in the Body Clock and other components of the Health Octo Tool.

There are limitations to this study. The BLSA study recruited primarily initially healthy individuals, focusing on healthy aging. Some of the inter-population differences can be due to the initial study recruitment inclusion criteria. Therefore, the general population, which may have more severe conditions and hereditary diseases, might experience a more accelerated Body Clock in comparison to chronological age or time. Nevertheless, since the Body Clock is quantified at the individual level, it can be updated whenever new personalized information on disease levels becomes available using our prediction models.

In the NHANES dataset, only mortality was considered as a longitudinal outcome, and we used the Body Clock to forecast mortality. It is important to acknowledge, however, that the NHANES data slightly underestimates the Body Clock due to the lack of information on specific diseases, such as thyroid conditions or macular degeneration, or ejection fraction from echocardiography in certain cohorts. Despite this constraint, the AUC for mortality prediction in this cohort remained at 0.8, surpassing the performance of the FI.

In summary, together, BSC, BSA, Body Clock, Body Age, Speed-Body Clock, Speed-Body Age, Disability-Body Clock, and Disability-Body Age comprise the Health Octo Tool—a multidimensional, comprehensive health assessment instrument. This marks the first instance of framing multimorbidity in terms of intrinsic clocks and their translation to rates of aging. Such metrics enable a deeper understanding of the genomic and environmental factors underlying heterogeneity in various aspects of health at any age. This tool can shed light on the variability of the biological aging and has the potential to differentiate personalized markers of comprehensive health, encompassing genomics and environmental factors. It can also identify organs and individuals with accelerated aging or at risk, and can be applied in all-age clinics, precision medicine, and clinical trials. By understanding the complex interplay between organ health, aging, and disease, this tool can contribute to improving healthcare strategies and identifying early-onset requirements for personalized interventions.

## Methods

This study employs secondary data analyses using data from three distinct sources: the Baltimore Longitudinal Study on Aging (BLSA)[31,32] [$n = 907$, women=451], the longitudinal Invecchiare in Chianti (InCHIANTI) aging study[33] [$n = 986$, women=551], and NHANES data (2003–2018)[38] [$n = 40,700$, women = 23,121]. These studies are summarized in the supplemental materials. This study was conducted in accordance with all relevant ethical regulations. The original BLSA and InCHIANTI studies were approved by the Institutional Review Board at the Intramural Research program at National Institute on Aging (NIA/IRP). The study participants provided informed consent for the future use of the data. Our analyses used de-identified data. The proposals using BLSA and InCHIANTI data for this project were approved at NIA/IRP, and NHANES data is publicly available.

### Bayesian inference

We briefly describe the formula for the Bayesian approach in the Supplemental Method. Bayesian inference allowed us to estimate the coefficient distribution around the observed point and build models for all points using both prior knowledge and likelihood. In our analyses, we used BODN as an ordinal longitudinal outcome with the cumulative family.

Briefly, the cumulative model operates under the assumption that the observed ordinal variable $Y$ is derived from the categorization of an underlying continuous latent variable $\widetilde{y}$. In this framework, there are latent thresholds $\tau_K$ (where $1 \leq \kappa \leq K$) that divide the continuous latent variable $\widetilde{y}$ into $K + 1$ distinct, ordered categories, which correspond to the observed ordered values of $Y$ (equation [18]).

$$Y = K, \ \tau_{K-1} < \widetilde{y} < \tau_K \qquad (18)$$

In this formula $K$ is 13, the number of subsequent values of BODN, and $\widetilde{y}$ is the predicted BODN in the model[28,30].

The disease levels are included in the models as lagged ordinal predictors, adjusting for time and excluding chronological age. To develop various models, we employed Multilevel Ordinal Regression with individuals as the model level within a Bayesian framework[28,30]. To assess the models' predictive accuracy, we used leave-one-out cross-validation[36], calculated the models' weights[37], and compared all models with chronological age. By incorporating all disease levels into a single model, we were able to predict post-analyses BODN at individual levels, quantifying an individual-based Body Clock. To determine Body Age, we regressed chronological age over the Body Clock with a Gamma distribution and log link.

### Latent maximum effect

To avoid crude categorization of diseases and the risk of disease misclassification in older adults, in addition to the ordinal BODN, we used specific diseases as ordinal variables within the Bayesian framework[29]. This approach treated ordinal categorized diseases as a continuous monotonic latent variable, which allowed us to estimate maximum coefficient effects for each ordinal predictor, referred to as maximum disease-specific posterior coefficient estimates. We then introduced ordinal cut-points based on the proportion of each disease level in the data to quantify disease-level-specific estimates.

### Model evaluation and in-sample and out-of-sample predictive checks

To assess the robustness of the model's ability to predict BODN, we employed the Time-1 full model based on BLSA data to predict BODN in the InCHIANTI and NHANES data for out-of-sample validation, and we used InCHIANTI and NHANES data as separate models for replication. We compared the estimated log predictive density (ELPD) of the models using LOO-CV, which allows us to compare the predictive distribution to the true data generating process. Additionally, we used posterior predictive checking[54] to compare the posterior predictive density of simulated data to density estimates of the observed data to predict the new data (for InCHIANTI and NHANES data, representing out-of-sample validations).

### Bayesian stacking weights to compare models

To compare several models simultaneously and determine their respective strengths, we used Bayesian stacking[53]. This method allowed us to compute model weights for each type of model, paired with their corresponding age-only models (single-disease, single-system, stepwise multisystem, and full models separately). Bayesian stacking optimizes the estimated LOO-CV predictive performance of the weighted model combinations, providing model-specific weights compared to the age-only models, guiding us towards the models with the best predictive performance.

### Reporting summary

Further information on research design is available in the Nature Portfolio Reporting Summary linked to this article.

## Data availability

The BLSA and InCHAINTI data are available under restricted access, because the elderly individuals are identifiable based on their age. The access can be obtained by submission of a proposal to the BLSA and InCHIANTI study research committees. NHANES data is available online through the link: https://www.cdc.gov/nchs/nhanes/index.html.

## Code availability

The codes for algorithms are presented in the GitHub and Zenodo pages[55]. https://github.com/ssalimi/HealthOctoTool https://zenodo.org/records/14835216.

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

## Acknowledgements

ShS is funded by National Institute on Aging K01 AG059898. DR is funded by Nathan Shock Center at University of Washington P30AZ013280. We are grateful to the participants of the BLSA and InCHIANTI, and NHANES studies, researchers, staff, and administrators. ShS acknowledges the Stan Community, and the Bayesian Data Analysis course targeted for Global South taught by AV. We appreciate Jefferey Laubenstein for painting Fig. S1.

## Author contributions

Sh.S. conceptualized the Body Clock, Body Age, Bodily System-Specific Clock, Bodily System-Specific Age, Speed-Body Clock, Speed-Body Age, Disability Index, Disability-Body Clock, and Disability-Body Age. Sh.S. managed the data, conducted statistical analyses, designed graphs, and authored the manuscript. Sh.S. designed the Figs. 1 and S1. A.V. supervised Bayesian analyses, advised on methods, and communicated statistical findings. M.S. contributed to interpreting multimorbidity indices, writing, and editing. D.R. edited the manuscript and provided feedback on aging rates, results, and discussion, and designed Fig. 1 with Sh.S. M.K. provided feedback on aging rates. F.L. contributed to health concepts, results interpretation, manuscript writing and editing, rates of aging, and insight on study methods in the InCHIANTI and BLSA data.

## Competing interests

The Health Octo Tool has a provisional patent (patent pending) by Sh.S. with the aim to make it digitally available to researchers. M.K. is a co-founder and shareholder of Optispan, Inc., and has expressed no conflict of interest with this work. The remaining authors declare no competing interests. This material should not be interpreted as representing the viewpoint of the US Department of Health and Human Services, the National Institutes of Health, or its represented agencies.
