## [Transparent Peer Review file · Nature Communications]

Health Octo Tool Matches Personalized Health with Rate of Aging

Corresponding Author: Dr Shabnam Salimi

Version 0:

Reviewer comments:

Reviewer #1

(Remarks to the Author)

Summary

In “Body Clock: Matching Personalized Multimorbidity and Fast Aging Using Information Entropy” Salimi and colleagues, in collaboration with Prof Luigi Ferrucci, introduce a new multimorbidity measure and present a statistical model of aging to offer a “proxy of the personalized rate of biological aging”. The data come from the community-based Baltimore Longitudinal Study on Aging (BLSA), a volunteer cohort study established in an era in which it was popular to distinguish ageing from illness and to understand what happens with healthy ageing.

The paper introduces the Body Organ Disease Number (BODN), calculated as the degree of impairment / presences of abnormalities in disease states for 11 organ systems as well as two distinct diseases (stroke and adult-onset cancer – they propose two rationales for counting these separately). The ~80 items that make up the BODN range from laboratory or other test abnormalities to clinical diagnoses. These were coded as deficits to some degree if they deviated from normal at a younger age, or met established criteria for disease. These are combined in a complicated way, using Bayesian regression to account for intervals in ordinal measures having varying widths.

The Body Clock being the exposure, several outcomes were considered, including Short Battery of Physical Performance (SBPP), the presence of so-called “geriatric syndromes” (“two injurious falls in past month, urinary or bowel incontinence, disability and mortality”) and new onset disability, defined as two incident IADL changes and/or one BADL change. Quite usefully, the analysis considers the prior state through introducing two body clock measures at prior states, thereby allowing for the heterogeneity of health deficit accumulation across the life course to be evaluated.

The paper demonstrates that this method is useful in predicting mortality and the other adverse outcomes considered.

Major comments

This paper has much to recommend it. It comes from a well-established cohort, and has been cross-validated. Over against the “geroscience agenda” of needing of study ageing to advance our understanding of the diseases of old age – only then often to tackle single “ageing mechanisms” - this method allows for the complexity of ageing to be addressed quantitatively. If “the wolves of old age hunt in packs” – this offers a means to understand how large a pack any older person faces – i.e. many problems an individual has at any age and how that itself might influence age-related disease. It also takes into account that where people land up often is determined by from whence they started. On the other hand, it is coming at this from a somewhat agnostic way, not building on useful prior work. Still, the method itself is not exactly transparent, and requires more prior information than would ordinarily be available. In consequence, it bears the burden of needing to be generally useful if its ambitious goal is to be realised.

1. The paper treads the well-trod and not unmarked ground of introducing a measure of “the personalized rate of biological aging”. The fruitful trail less often followed is, as was also not done here, has the new measure compared with predecessors. AT such a high-impact journal, there is no reason to wait a year or two until this group or others take up the cudgels to see how it compares. This can be done now, in my view.

a. Two 2020 papers in this regard. One, of this parish, comes from the Sinclair lab: it introduced two CLOCK methods (one for mortality, one for estimating the efficacy of any lifespan-extending intervention, each used machine learning to evaluate separate components of the frailty index); each was compared with the frailty index as a whole. Schultz MB, Kane AE,

Mitchell SJ, MacArthur MR, Warner E, Vogel DS, et al., Age and life expectancy clocks based on machine learning analysis of mouse frailty. *Nat Commun.* 2020 Sep 15;11(1):4618. doi: 10.1038/s41467-020-18446-0. Erratum in: *Nat Commun.* 2020 Oct 8;11(1):5143. PMID: 32934233; PMCID: PMC7492249.)

b. The second paper, an international collaboration led from the Karolinska, compared nine measures – all of which predicted mortality. It then explored which measures worked best, concluding that “in joint models, two methylation age estimators... and FI remained predictive, suggesting they are complementary in predicting mortality”. The paper also lays out some useful starting points for any measure of biological age: it “should be able to predict the risk of individuals in the group dying during the follow-up period – as each did”; it should demonstrate -as many have done - that biological aging appears to accelerate after age 65 years, and that there are consistent sex differences.”

c. In short, the paper demonstrates that it can do what others have done (in this regard note especially Stolz E, Hoogendijk EO, Mayerl H, Freidl W. Frailty changes predict mortality in four longitudinal studies of aging. *J Gerontol A Biol Sci Med Sci.* 2020 Oct 26;glaa266. doi: 10.1093/gerona/glaa266. Epub ahead of print. PMID: 33103718. And the v recent Hoogendijk EO, Stolz E, Oude Voshaar RC, Deeg DJH, Huisman M, Jeurink HW. Trends in Frailty and its Association with Mortality: Results From the Longitudinal Aging Study Amsterdam (1995-2016). *Am J Epidemiol.* 2021 Feb 4;kwab018. doi: 10.1093/aje/kwab018. Epub ahead of print. PMID: 33534876. But can it do it better? What value is added by the more complicated approach on offer here?

2. Little of the data are presented by sex. We only have that for one Figure, where detectable differences are present. In the spirit of NIH policy, if sex differences be present, they should be better acknowledged than the scarce treatment now accorded in this paper. [Clayton JA, Collins FS. Policy: NIH to balance sex in cell and animal studies. *Nature.* 2014 May 15;509(7500):282-3. doi: 10.1038/509282a. PMID: 24834516; PMCID: PMC5101948.]

Other comments.

1. Figure 1 helps to illustrate the notion of multiple levels of age-related health deficits were considered across several body systems and with a range of scale. Even so, runs the risk of looking more like advertising than anything else.
2. One interesting consequences of this approach is that it appears to capture change only in the direction of worsening. How would improvement in health be captured?
3. Why in the CBC no WBC?
4. No delirium in the list of “geriatric syndromes”?

Reviewer #2

(Remarks to the Author)

This article reports on a novel statistical approach to measure the construct of the rate of aging. This rate is thought to determine the timing of onset of disability as well as lifespan. The authors conduct analysis of 2-time-point repeated measures clinical data in a long-running sample of adults. They devise an algorithm from these data to measure the rate of aging. They show this algorithm is associated with physical performance deficits, disability, and mortality.

The main contribution of this paper is to propose a more statistically sophisticated, longitudinal-data-based version of what in gerontology is referred to as a frailty index. Frailty indices attempt to summarize the overall state of health in an individual by counting up health deficits across different systems in the body. The innovation in this paper is the combination of frailty method with a Bayesian statistical framework and longitudinal data. In this way, the method is sort of a mashup of the standard frailty model with the pace of aging model introduced by Belsky and colleagues. Methods that focus on rate of change within individuals are an important complement to methods focused on states observed at a cross-section. So there is potential in the approach the authors propose.

However, no data are provided to establish any value added of the novel method, which is far more complicated than a traditional frailty index. It is already known that people who have more diseases are more likely to become disabled and to die. Having a higher incidence rate of disease may be assumed to do the same. What needs to be established is that this rate of increase in disease burden measured by this new method adds value to tools we already have. To do this, the authors need to (a) compare their method to standard frailty indices within their own data; (b) apply the algorithm derived within the training data to a novel dataset to establish external validity beyond their training sample; and (c) repeat the comparisons of their method to others in at least one additional dataset.

Reviewer #3

(Remarks to the Author)

The goal of the reviewed study was to create and validate the "body clock" number, as instrumentalized using an enumeration of the number of bodily systems with the presence of diagnoses consistent with a systemic dysregulation approach. The ostensible goal of this effort is to create an "aging" index or, if not that, of creating a new measure on which to rely as a proxy for biological (non-chronological) age. The goal is reasonable, and the theory is generally well formulated. The main issues are 1) implementation and 2) competition.

In terms of implementation, I found the paper to be somewhat lacking. Notably, the validation data were opaque and not clearly targeted at answering the researcher's questions. Additionally, many of the datapoints were used as validation but seemed like they were tautological. For example, using the predictive power of an enumeration of the number of systems (or types of systems) dysregulated to predict the body clock number, which is designed on those very same predictors. This seemed like a waste of time, and was confusing to see. Nevertheless, there were interesting points of information in those analyses showing, for example, that heart failure had a worse overall clinical course as compared to other diseases.

A second concern with implementation is the reliance on clinical diagnoses rather than research-based criteria for measurement. There are a huge range of reasons that a specific individual in a study might not be able to get a diagnosis for a condition including that they live in a place where diagnoses are hard to come by, or where diagnosticians are not very good. In such cases, it would be worthwhile to have biomarkers or symptom scores to rely on that specifically targeted each of these systems and allowed researchers to collect standardized information for comparison across cultures and healthcare systems. As it is, however, it is likely that people living in high-service environments will appear to be aging more rapidly than those with poorer access to healthcare. But we know that this is not because those with worse access are aging more rapidly on the whole.

I found the literature review somewhat limited, and potentially problematically so, since there are a proliferation of these types of efforts. It would be useful to know, for example, how this paper expands information available from other biological age calculators. For example, GrimAge (Lu et al. 2019) is an epigenetic clock that has recently emerged that is quite interesting, but perhaps more compelling to this paper is its correspondence/deviation from the Cambridge Multimorbidity Score, which seems to be calculated in a similar overall way but appears to be better validated (Payne et al. 2020).

Minor issues:

The first "c" in the manuscript is typed in bold face.

The "SPPB" is mislabelled in the tables as SBPP.

Version 2:

Reviewer comments:

Reviewer #1

(Remarks to the Author)

(Remarks on code availability)

Reviewer #2

(Remarks to the Author)

Please see attached do for a version of this that is better formatted.

The authors have mostly addressed my initial comments. The additional data substantially strengthen the paper, adding confidence that the method illustrated generalizes beyond the cohort in which it was developed. However, I think the core analyses establishing clock predictions of important outcomes could be better represented in the figure set, which largely reports details of the clock, but lacks compelling visualization of evidence that the clock matters. I also think that A LOT of work is needed to improve clarity and readability.

1. I am still missing analysis comparing the new body clock to existing frailty indices. The response letter says they are there. But I had trouble finding them. Maybe they could be reported in an easier to find place? Basically, it would be nice to see IRRs and HRs with their CIs for the various outcomes in the various datasets plotted side by side parallel effect-sizes estimated for a standard frailty index. This type of visualization is a mainstay of work in epigenetic clocks and is tremendously clarifying for readers re the robustness and value added of a new measure.

2. It would also be useful to see how the models of BODN (i.e. the regression weights estimated for different diseases in the model used to derive body clock) compare across the 3 datasets. Are they similar? It would seem to me that if BODN and Body Clock are meant to be *measures*, the approach would be to derive the Body Clock model in one dataset and then apply it in another. However, if instead this suite of tools is intended as a set of *methods* that can be used to discover properties within a dataset, then it would seem a key observation is how those discovered properties may vary or be conserved.

3. More generally, the paper could be made more user friendly. There are a lot of data being integrated with some pretty sophisticated methods. If there is real value add to prediction of adverse aging outcomes – or even if this novel approach just matches

standard FIs in terms of predictive power -- there is something new and important here. But some additional work is needed to make that contribution available to readers.

Below are some suggestions toward moving the paper in this direction:

a. There are SO many new measures in this paper -- Body Clock, BODN, Body Age, Bayesian Disability Index... There are a lot. I found it very hard to keep track of what each one was and why they mattered/what they meant in terms of the overall argument. I think the paper needs a text-box or figure that provides a glossary of these new measures that defines them and also explains their role in the analysis. A graphical-abstract-type figure could also accomplish this. I think figure 1 is suppose to do this job. It doesn't. In fact, the initialization-rich figure and the exceptionally terse legend only serve to the confuse the reader more. Most will be driven to despair (or just the next open browser tab) before they get a chance to appreciate what the authors are after.

b. The abstract is impossible to follow. I read it maybe a dozen times and couldn't figure it out. If a reader starts there, they will stay lost forever. In contrast, the opening paragraphs of the introduction are quite clear and make a compelling case for the new measure/suite of measures being introduced. Perhaps re-drafting the abstract focusing on the high-level goals of the project would be a better strategy.

c. Simpler language would go a long way toward readability. For example, on page 4, still in the introduction, the authors define their first novel measure, BODN, as "the number of organ systems with at least one deviation from health due to diseases or impairments, providing an unequidistant ordinal value so that the severity of diseases or disease levels can variably contribute into this unequidistant metric, and we can obtain their coefficients as the weight of their contribution into BODN using a Bayesian framework"

I have no idea what that means. I have high confidence that neither do 99% of the readers of Nat Aging. From the first paragraph of the results, I would be inclined to understand BODN as simply the count of organ systems with evidence of pathology. But that sentence suggests something much more exotic. And down in the results, after the apparently simple explanation as count of organs with pathology, the text returns to "nonequidistant..." Is the point that BODN is calculated as a count, but modeled as an ordinal variable to relax the assumption that each step up the count represents the same magnitude of increase in body entropy?

d. Being more clear earlier on about what data are used to define BODN would be helpful. There is a lot of talk about modeling BODN and BODN dynamics from disease data. But one wonders how disease data are distinct from the data used to calculate BODN. Is BODN computed from biomarkers? I couldn't find the details.

e. The same goes for body clock. The authors offer very little detail about how these measures are actually constructed, offering a single, ultra-dense sentence that makes no reference to the data being used, and then galloping off to describe the properties of the novel metric. The closer I looked, the less I understood.

MINOR

The authors should not abbreviate chronological age as "c-age" this is unnecessary and makes the paper harder to read/understand. There are enough new things in this paper that we don't need new names for things we already understand.

(Remarks on code availability)

Reviewer #3

(Remarks to the Author)

The authors have adequately addressed my comments and I have no further concerns.

(Remarks on code availability)

Version 3:

Reviewer comments:

Reviewer #2

(Remarks to the Author)

The authors have done a lot of work and substantially addressed one of my two major concerns. In my last review I flagged

needs for (1) comparative validation vs. frailty indices; and (2) clearer explanation of the method. I think the authors delivered on (1). They clearly made an effort on (2). But, in my view, they are not there yet – frankly, they are not close.

I do not understand what is at the core of this toolkit. I think the idea is that by modeling a count of organ systems with pathology as an ordinal variable from longitudinal data on individual diseases, the authors can generate a predicted value that represents something more sophisticated/nuanced than the original count. Everything else follows from that predicted value. But it's not clear from the paper as written what that predicted value is. It would be helpful (a) to state that clearly how BODN is measured (i.e. from what variables combined in what way taking on what resulting distribution of values); and (b) to do the same for the various models of BODN – i.e. what are the predictor variables? I think the underlying theory of measurement here is that modeling a multimorbidity index from longitudinal organ-system data can generate predicted values that provide a deeper summary of the aging process than the count itself. But honestly I'm just not sure.

I also can't tell if the tools reported in this paper are algorithms for which parameters are estimated in the training data (BLSA) and variable values are input from the test data (InCHIANTI, NHANES) or if instead they are methods that can be applied to new datasets to derive dataset-specific parameters. Both can be useful. But I am not clear on which one is being introduced here.

The authors need to be clear about what the measurements they analyze are and what the models they fit to those measurements include as covariates. They should also provide lists of the variables they are including in their analyses (e.g. in supplementary tables if necessary) and equations that show how the parameters they describe are estimated.

Finally, related to the above comment, this paper is presented as introducing a toolkit. But I can't tell what the tools are. Are there equations published in the paper that constitute the tools? (I didn't see any) Is there software that will be published with the article that will allow others to implement these tools in new data? If so, what inputs does that software require?

The validation analysis is a bit more clear – at least the evidence that the metrics computed in the analysis are predictive of stuff we care about and generally more so than standard frailty indices. But here too things are needlessly opaque. What diseases were counted to compute frailty?

SPECIFIC COMMENTS

Introduction para 3 (p4 para2) needs to be re-written. It's critical motivation for the paper, where the authors aim to problematize existing algorithm-based measurements of aging and highlight areas where the new approach adds value. They need to get it right. Here is what stood out to me in the reading:

The first sentence lumps together the Horvath, PhenoAge, GrimAge, and DunedinPACE epigenetic clocks as examples of approaches that use chronological age as a reference to predict risk for adverse health outcomes. As written, the sentence is accurate only about the Horvath clock (which is a rather poor predictor of adverse health outcomes). The other three clocks (which are much better predictors of adverse health outcomes) were developed using different methods. The model used to develop PhenoAge included chronological age as part of a composite reference phenotype. In the case of GrimAge, chronological age is actually a component of the algorithm (so not part of the reference at all). In the case of DunedinPACE, the reference is a composite measure of rates of change in multiple organ systems over 20y of follow-up and sample used to develop it was all the same chronological age (so again, no chronological age in the reference).

The second sentence makes the claim that chronological age was included in "equations to predict" various outcomes "as part of the indices". I'm not sure what this is meant to convey. But it again sounds like the researchers include chronological age in the "biological age" measurement they are using to predict outcomes. What is true of the papers cited is that they covary for chronological age in models predicting adverse health outcomes with the goal of isolating the unique information captured in the biomarker algorithm. That's quite a bit different from what the sentence implies. Also, the papers cited are about different measures than the ones introduced in the previous sentence.

The third sentence makes a claim about heteroskedasticity in prediction errors from biological age algorithms. While no citations are provided, I know this claim tends to be true of so-called "1st-generation" epigenetic clocks. That's a pretty limited subset of the space being covered in this para. It's also not an obvious limitation – we might assume that the oldest people look biologically younger than model expectations because, as suggested by their having lived longer than most of their birth cohort, their rate of aging is actually slower...

Fourth, the final sentence of the para concludes with a broad claim about algorithms developed using survival/mortality as a reference being vulnerable to external validity issues arising from the architecture of mortality varying over time and space. I am sympathetic to the idea being expressed here. But it's basically a non-sequitur in this paragraph. In the first case, the authors have not introduced such clocks previously. So the reader doesn't know what they are talking about. But more importantly, this is a big complicated idea and not one to be passed off in a single sentence supported by a lone ref that is mostly about a different topic. I also note that the validation evidence for clocks trained to predict mortality (Grim, Pheno) is pretty robust across settings with significant differences in mortality architecture.

In sum, this is a bad paragraph from top to bottom. The authors should scrap it and start over. They need an argument for what their new measures add to the literature. This isn't it.

For my taste, the next paragraph is equally confused and hard to follow. There is a good argument to be made for this tool. But it has to be made. I cannot follow what is written in this paragraph and the referencing doesn't help – many of the papers don't seem to be relevant to the claims being made...

I would suggest scrapping the discussion of biological age metrics altogether. They are difficult to explain and it's not clear to me that this toolkit is intended as a competitor to them. Rather, as articulated in the opening paragraphs, which I liked, it sounds like the goal here is a more sophisticated version of a multimorbidity index. I would focus the argument for the

knowledge gap addressed/value added on that.

MINOR: Use of "c-age" to abbreviate chronological age is unnecessarily hard on the reader. Just spell it out.

I have not commented on the discussion because I don't feel I understand what comes before sufficiently well to do so.

(Remarks on code availability)

Version 4:

Reviewer comments:

Reviewer #3

(Remarks to the Author)

To the Authors -

This review response is very good, and I think it responds to many of the prior concerns. I still see the novelty of the work, though I do think there are some chances for editorial improvement even now. I have a few suggestions, below. However, the methodological choices and results seem quite robust and so I focused more on those findings for my scoring here.

1) The spaghetti plots could benefit from having an average overlaid. It looks to me as though several of the relationships indicate the presence of accelerated change in older ages, which I think are based on the monotonicity assumptions of the model, so it would be good to allow some curvature to the line of best fit.

2) I'm not sure I see the benefit of this discussion paragraph: "Using Body Clock regressed over chronological age, we determined rate of entropy age as Body Age. Furthermore, Body Age serves as a key link between health entropy and rate of aging at the individual level." The one that follows two after is also short and similarly unhelpful.

3) The abstract shouldn't use the word "effect" and should also contain critical datapoints like accuracy and predictive power that are presented in the manuscript and would help to convince the reader of the manuscript's utility.

4) The abstract says: "Moreover, aging is a multidimensional process which cannot be captured by a single metric. Therefore, we assessed global health examining disease ... ". This is verbose and should be: ""Since aging is a multidimensional process that cannot be captured by a single metric, we assessed systemic health by examining disease ... ""

5) The sunburst graphs are in their native forms, but this is unhelpful and looks odd in my version. Could you remove the checkbox with "Legend" next to it and also maybe put all of them on a white background? If possible to make them bigger, that could also help as they are pretty small on my page and difficult to read.

(Remarks on code availability)

The code is relatively simple for this kind of code and it is well enough justified. The readme file looks good and I understood how to use the code, though I do think that it is only usable for a person with a strong background in R or Python.

REVIEWER COMMENTS

Reviewer #1 (Remarks to the Author):

Summary

In “Body Clock: Matching Personalized Multimorbidity and Fast Aging Using Information Entropy” Salimi and colleagues, in collaboration with Prof Luigi Ferrucci, introduce a new multimorbidity measure and present a statistical model of aging to offer a “proxy of the personalized rate of biological aging”. The data come from the community-based Baltimore Longitudinal Study on Aging (BLSA), a volunteer cohort study established in an era in which it was popular to distinguish ageing from illness and to understand what happens with healthy ageing.

The paper introduces the Body Organ Disease Number (BODN), calculated as the degree of impairment / presences of abnormalities in disease states for 11 organ systems as well as two distinct diseases (stroke and adult-onset cancer – they propose two rationales for counting these separately). The ~80 items that make up the BODN range from laboratory or other test abnormalities to clinical diagnoses. These were coded as deficits to some degree if they deviated from normal at a younger age, or met established criteria for disease. These are combined in a complicated way, using Bayesian regression to account for intervals in ordinal measures having varying widths.

The Body Clock being the exposure, several outcomes were considered, including Short Battery of Physical Performance (SBPP), the presence of so-called “geriatric syndromes” (“two injurious falls in past month, urinary or bowel incontinence, disability and mortality”) and new onset disability, defined as two incident IADL changes and/or one BADL change. Quite usefully, the analysis considers the prior state through introducing two body clock measures at prior states, thereby allowing for the heterogeneity of health deficit accumulation across the life course to be evaluated.

The paper demonstrates that this method is useful in predicting mortality and the other adverse outcomes considered.

Major comments

This paper has much to recommend it. It comes from a well-established cohort, and has been cross-validated. Over against the “geroscience agenda” of needing of study ageing to advance our understanding of the diseases of old age – only then often to tackle single “ageing mechanisms” - this method allows for the complexity of ageing to be addressed quantitatively. If “the wolves of old age hunt in packs” – this offers a means to understand how large a pack any older person faces – i.e. many problems an individual has at any age and how that itself might influence age-related disease. It also takes into account that where people land up often is determined by from when they started. On the other hand, it is coming at this from a somewhat agnostic way, not building on useful prior work. Still, the method itself is not exactly transparent, and requires more prior information than would ordinarily be available. In consequence, it bears the burden of needing to be generally useful if its ambitious goal is to be realized.

1. The paper treads the well-trod and not unmarked ground of introducing a measure of “the personalized rate of biological aging”. The fruitful trail less often followed is, as was also not done here, has the new measure compared with predecessors. AT such a high-impact journal, there is no reason to wait a year or two until this group or others take up the cudgels to see how it compares. This can be done now, in my view.

Reply. First, we would like to thank the reviewer for the insightful and encouraging comments that provided us with the opportunity to further elaborate the approach we have taken in developing the BODN and Body Clock. We highlighted the changes in the changes in the manuscript and supplemental information.

We have now extended the BODN and Body Clock to eight metrics representing several dimensions of health and rate of aging in the BLSA cohort and used the InCHIANTI study and NHANES data for validation. We modified the title of the manuscript accordingly to

“Body Clock as a Roadmap to Health Octo Tool: Matching Personalized Health and Multimorbidity with Rate of Aging.” **Page 1 title.** We also updated the Abstract accordingly.

As a predecessor metric, the frailty index algorithm has been the main metric that has some similarity to our index. Although the initial Canadian Frailty Index score (FI) was applied to cross-sectional data of individuals aged 70 and older, it has evolved since then, as various FIs using clinical data and other biomarkers have been introduced. Therefore, in our study, we chose to compare our algorithm with a FI that includes the same diseases used in our study, as suggested by the three reviewers and the editor. Below we point out our findings and discuss them in response to the esteemed reviewers and the editor.

We developed Bayesian Disability Index comprises of 47 items using Zero Inflated Beta Binomial as late-onset outcomes to be predicted. **(Highlighted Manuscript: Introduction, Page 5, Results: Figure 1 a, Page 7, Page 14-15, Supplemental Info: Page 9, Table S2, Page 14).** We then compared the Body Clock's performance with the manually developed FIs using a Bayesian approach and leave-one-out (LOO-CV). Our results demonstrated that the Body Clock significantly predicted DI, displaying a coefficient of 0.32 (95% CI=0.28–0.36). Model comparisons indicated that Body Clock exhibited better model performance comparing “expected log pointwise predictive density” (ELPD) which showed a larger ELPD, for Body Clock compared to FIs, and the difference in ELPD, favoring DI with the larger ELPD in both BLSA and InCHIANTI data. **(Supplemental Info: Page 30)**

Of note, the initial FI was developed in a population with ages 70 years and older and the items included in the FIs were mainly late-onset symptoms/diseases such that the FI score (FIs) tracked well with chronological age. When individuals younger than 70 years or more diseases are included in the FIS, however, the FIS score is mainly skewed toward older age, overlooking the process of aging might start earlier in life at sub-clinical stage and long before development of late-onset outcomes after age 65 years and older.

We also developed Disability-Body Clock and their pertained rate of aging (Disability-Body Age to capture functional and cognitive resilience in presence of multimorbidity in both the BLSA and InCHIANTI data **(Highlighted Manuscript: Page 14-15).**

These metrics offer a nuanced understanding of the transition between aging and pathology and provide a foundation for comprehensively assessing health across the life span.

- a. Two 2020 papers in this regard. One, of this parish, comes from the Sinclair lab: it introduced two CLOCK methods (one for mortality, one for estimating the efficacy of any lifespan-extending intervention, each used machine learning to evaluate separate components of the frailty index); each was compared with the frailty index as a whole. Schultz MB, Kane AE, Mitchell SJ, MacArthur MR, Warner E, Vogel DS, et al., Age and life expectancy clocks based on machine learning analysis of mouse frailty. *Nat Commun.* 2020 Sep 15;11(1):4618. doi: 10.1038/s41467-020-18446-0. Erratum in: *Nat Commun.* 2020 Oct 8;11(1):5143. PMID: 32934233; PMCID: PMC7492249.)
- b. The second paper, an international collaboration led from the Karolinska, compared nine measures – all of which predicted mortality. It then explored which measures worked best, consuming that “in joint models, two methylation age estimators... and FI remained predictive, suggesting they are complementary in predicting mortality”. The paper also lays out some useful starting points for any measure of biological age: it “should be able to predict the risk of individuals in the group dying during the follow-up period – as each did”; it should demonstrate -as many have done - that biological aging appears to accelerate after age 65 years, and that there are consistent sex differences.”
- c. In short, the paper demonstrates that it can do what others have done (in this regard note especially Stolz E, Hoogendijk EO, Mayerl H, Freidl W. Frailty changes predict mortality in four longitudinal studies of aging. *J Gerontol A Biol Sci Med Sci.* 2020 Oct 26:glaa266. doi: 10.1093/gerona/glaa266. Epub ahead of print. PMID: 33103718. And the v recent Hoogendijk EO, Stolz E, Oude Voshaar RC, Deeg DJH, Huisman M, Jeuring HW. Trends in Frailty and its Association with Mortality: Results From the Longitudinal Aging Study Amsterdam (1995-2016). *Am J Epidemiol.* 2021 Feb 4:kwab018. doi: 10.1093/aje/kwab018. Epub ahead of print. PMID: 33534876. But can it do it better? What value is added by the more complicated approach on offer here?

We appreciate and acknowledge the insightful comments, suggestions and refereeing to the valuable published articles. We cited these articles these articles as developing metrics based on mortality prediction or developing FI including late-onset outcomes such as items of disability and cognition deficit to predict mortality which is mainly either included people aged 65 years (Stolz et al.) or older or skewed toward age 70 years old (Li *et al.* PMID: 32041686). Of note, all of these studies predicted mortality and mainly FI was developed in older adults including late-onset outcomes. Interestingly Hoogendijk *et al.* (the paper that the reviewer mentioned) showed that recent year’s frailty increases without change in mortality revealing the need to address health changes decoupled from mortality. The Body Clock predicts mortality, functional decline, SPPB and disability and now we showed that it predicts walking speed and DI. **(Highlighted Manuscript Page 13-15)** As we mentioned above, in our view the caveat of developing a frailty index by including all late-onset phenotypes makes the prediction model skewed toward old age while under-estimating early-onset processes. Our concern is to capture early-onset process of aging and predict functional outcomes such as walking speed, SPPB, and disability for enhancing healthspan. The complexity of our Bayesian approach allowed us to use sub-clinical and clinical measures to capture both health and the rate of aging and thus provide a clinical understanding what the rate of aging means in terms of health. Our approach allowed us to require fewer markers such as methylation to be included. Moreover, our metrics can be expanded and used in the future to understand epigenetics and genetics as well as environmental factors. We also used the same data and information to extend BODN and Body Clock to new additional metrics as components of the Health Octo Tool independent of mortality, chronological age or late-onset outcomes so that we can show organ-system clocks and aging, functional aging and disability aging **(Highlighted Manuscript Page 7, Page13-15, Table 1, Table S2).**

Even at very old age a subclinical aging process contributes to longitudinal BODN and predicts late-onset outcomes. The recent preprint study of Belsky (long after we submitted this manuscript as a preprint and to the journal) showed that the DUNDONIAN methylation markers predict each of functional outcomes although only in white people, which is quite insightful. It still not known

how population heterogeneity play role in these metrics. Now, we believe that such markers can be used to assess and predict phenotype of aging in terms of Health Octo Tool as a multi-dimensional health assessment. Additionally, we have now developed **Body Age** that captures rate of aging, and we showed a heterogeneity of aging in both organs and at the individual level (**Highlighted Manuscript Page 12-13, Figure 5, Supplemental Info: Figure S5 and S6**). Additionally, as part of Health Octo Tool, we now showed that estimated aging in some organs can surpass reported chronological age, and this novel finding suggests future genetics and epigenetic studies may be able to help determine underlying mechanisms (**Highlighted Manuscript: Page 9, Figure 3a-c**).

Another crucial aspect of the Body Clock and Body Age are that these metrics incorporate the body entropy so that the effect of each disease level on all organs for each person, individually, can be calculated. Our metrics demonstrate that the effect of aging is extremely heterogeneous in terms of the age of onset and, therefore, it cannot be solely captured with FIs at early age.

The model showed reliability and was replicable in InCHIANTI study and NHANES data. It can be used to include new information on a new disease entity, predict a new person's Clock using a Bayesian approach, and provide the uncertainty in the prediction. We added references suggested by the reviewer (**Manuscript: Added InCHIANTI results throughout the results sections**).

2. Little of the data are presented by sex. We only have that for one Figure, where detectable differences are present. In the spirit of NIH policy, if sex differences be present, they should be better acknowledged than the scare treatment now accorded in this paper. [Clayton JA, Collins FS. Policy: NIH to balance sex in cell and animal studies. Nature. 2014 May 15;509(7500):282-3. doi: 10.1038/509282a. PMID: 24834516; PMCID: PMC5101948.]

We appreciate the reviewers' comments. We have now performed sex-difference analyses for the Body Clock, Speed Body Clock, Disability-Body Clock and Disability Index and reported the results in the BLSA and InCHIANTI data. In most studies the Body Clock was larger in women than men, as well as Speed-Body Clock and Disability-Body Clock were poorer in women than men (**Highlighted Manuscript: Results sections, Page 12-15 for BLSA and InCHIANTI study, and Page 16-17 for NHANES data**).

Other comments.

1. Figure 1 helps to illustrate the notion of multiple levels of age-related health deficits were considered across several body systems and with a range of scale. Even so, runs the risk of looking more like advertising than anything else.

We appreciate the reviewers' comment. We moved the figure to the supplemental materials.

2. One interesting consequences of this approach is that it appears to capture change only in the direction of worsening. How would improvement in health be captured?

We appreciate the reviewer's comment. The Body Clock is positive as the association of the most diseases with BODN are positive (**Figure 4 a&b, Table S4**). Improvements can be captured by decreases in the magnitude of the Bodily System Clocks, Body Clock, Disability-Body Clock, and decreases in Bodily System Age, Body Age, Disability-Body Age and decreases in Disability Index, as well as and increases in Walking Speed along with decreases in the Body Clock.

3. Why in the CBC no WBC?

We appreciate the recommendation. We included WBC as normal, leukopenia (low WBC count) and leukocytosis (high WBC count) into the models, and updated all analyses, and figures (**Figure 4, Supplemental Info: Table S1**).

4. No delirium in the list of "geriatric syndromes"?

Although delirium is an important Geriatric Syndrome, it is very difficult to capture delirium in the context of an epidemiological study as it is a rather hospital-based and an acute-onset phenomenon. People who have episodes of delirium do not report it in epidemiological studies or it is limited, and delirium is seldom mentioned in administrative data. Indeed, it would have been interesting to look at delirium, but this information is not in BLSA nor the InCHIANTI data. However, our algorithms can incorporate new information and be updated whenever new health information is available.

Reviewer #2 (Remarks to the Author):

This article reports on a novel statistical approach to measure the construct of the rate of aging. This rate is thought to determine the timing of onset of disability as well as lifespan. The authors conduct analysis of 2-time-point repeated measures clinical data in a long-running sample of adults. They devise an algorithm from these data to measure the rate of aging. They show this algorithm is associated with physical performance deficits, disability, and mortality.

The main contribution of this paper is to propose a more statistically sophisticated, longitudinal-data-based version of what in gerontology is referred to as a frailty index. Frailty indices attempt to summarize the overall state of health in an individual by counting up health deficits across different systems in the body. The innovation in this paper is the combination of frailty method with a Bayesian statistical framework and longitudinal data. In this way, the method is sort of a mashup of the standard frailty model with the pace of aging model introduced by Belsky and colleagues. Methods that focus on rate of change within individuals are an important complement to methods focused on states observed at a cross-section. So there is potential in the approach the authors propose.

However, no data are provided to establish any value added of the novel method, which is far more complicated than a traditional frailty index. It is already known that people who have more diseases are more likely to become disabled and to die. Having a higher incidence rate of disease may be assumed to do the same. What needs to be established is that this rate of increase in disease burden measured by this new method adds value to tools we already have. To do this, the authors need to (a) compare their method to standard frailty indices within their own data; (b) apply the algorithm derived within the training data to a novel dataset to establish external validity beyond their training sample; and (c) repeat the comparisons of their method to others in at least one additional dataset.

Thanks to the reviewer for the remarkable points and suggestions.

aWe believe in response to the reviewer 1 all three points are addressed. For point (a) we developed Frailty Index (FI) using the same items of Body Clock and compared Body Clock and FI to predict disability as an outcome (b) We replicated Body Clock the in the InCHIANT and NHANES data. (c) We compared the FI methods in BLSA and InCHIANTI studies, and in both databases, we found that including the disease levels predicted outcomes better than FIs. We developed the Bayesian-Based Disability Index that can be used as a novel outcome without instability due to zero values by applying a negative Beta Binomial Distribution approach. **(Highlighted Manuscript Page 14-15).**

Reviewer #3 (Remarks to the Author):

The goal of the reviewed study was to create and validate the "body clock" number, as instrumentalized using an enumeration of the number of bodily systems with the presence of diagnoses consistent with a systemic dysregulation approach. The ostensible goal of this effort is to create an "aging" index or, if not that, of creating a new measure on which to rely as a proxy for biological (non-chronological) age. The goal is reasonable, and the theory is generally well formulated. The main issues are 1) implementation and 2) competition.

In terms of implementation, I found the paper to be somewhat lacking. Notably, the validation data were opaque and not clearly targeted at answering the researcher's questions. Additionally, many of the datapoints were used as validation but seemed like they were tautological. For example, using the predictive power of an enumeration of the number of systems (or types of systems) dysregulated to predict the body clock number, which is designed on those very same predictors. This seemed like a waste of time and was confusing to see. Nevertheless, there were interesting points of information in those analyses showing, for example, that heart failure had a worse overall clinical course as compared to other diseases.

We thank the reviewer for the valuable comment. We added and further elaborated on the implementation of the algorithm in the Discussion section. **(Highlighted in the Manuscript/Discussion Section, Page 20).**

We compared the algorithm with FI and validated it in two more studies, the InCHIANTI and NHANES data sets.

To address the concern of the reviewer as: "For example, using the predictive power of an

enumeration of the number of systems (or types of systems) dysregulated to predict the body clock number, which is designed on those very same predictors”

The cumulative statistic family's ordinal value, as highlighted in the cited Bayesian paper, isn't uniformly spaced. Organ diseases, serving as lagged predictors for future BODN contribute unevenly as shown in varying coefficients. This unevenness results from the heterogeneity in the disease level's impact in terms of the magnitude of their influence on BODN.

We also added the following text to the manuscript (**Highlighted Manuscript, Page: 4, Page 6**).

“BODN quantifies the number of organ systems with at least one deviation from health due to diseases or impairments, providing an unequidistant ordinal value so that the severity of diseases or disease levels can variably contribute into this unequidistant metric, and we can obtain their coefficient as the weight of their contribution into BODN using Bayesian framework.”

“It is treated as an ordinal outcome so that numbers are not necessarily equidistant, enabling quantifying the variable contributions of diseases severity or levels to this unequidistant value³³. Furthermore, ordinal numbers were assigned to the disease levels representing the severity of each organ-specific health and disease, serving as lagged contributors of BODN. This approach allows for the detained and nuanced quantification of their effect and impact on BODN levels³⁴. We performed longitudinal data analyses from two studies: the Baltimore Longitudinal Study of Aging (BLSA)^{35, 36} and the Invecchiare in Chianti (InCHIANTI) study³⁷. Bayesian inference and Cumulative multilevel ordinal regression were employed to estimate posterior coefficient values for lagged diseases contributing to BODN^{33, 38-40}.”

Moreover, we would like to emphasize that the disease levels are not predictors of BODN but are incorporated into unequidistant values of BODN. Moreover, none of the disease levels showed any spurious associations in the results and the model performance and uncertainty checks all remained sound. This approach spares us the need to weight any single disease equal to 1 or arbitrary weigh them as 0, 0.25, 0.5, 0.75, and 1. The Body Clock is rather based on coefficient estimates that are incorporated into BODN. While in FI, if someone has congestive heart failure that is coded as a 1 and if they have cataracts that is also coded as 1. Both get the same proportional value, while we showed there are heterogeneities in the impact on BODN. We used the same disease levels to predict late-onset functional outcome and DI. We compared all these models for their predictivity performance. The Body Clock predicted late-onset outcomes and DI better than FI. (**Please see response to the reviewer 1**). Using a Bayesian approach allowed us to obtain individual-level metrics to use in the clinic and predict similar people with similar disease conditions for interventions not only to target the multimorbidity but also their function. By contrast, including all items as frailty does not allow us to disentangle which diseases are incorporated into the Body system or into functional health or resilience. Moreover, another goal of ours is to increase healthspan and disentangle those who have an increased Body Clock but also have functional resilience. These algorithms will help us to understand the common and distinguishing mechanisms of aging and resilience.

A second concern with implementation is the reliance on clinical diagnoses rather than research-based criteria for measurement. There are a huge range of reasons that a specific individual in a study might not be able to get a diagnosis for a condition including that they live in a place where diagnoses

are hard to come by, or where diagnosticians are not very good. In such cases, it would be worthwhile to have biomarkers or symptom scores to rely on that specifically targeted each of these systems and allowed researchers to collect standardized information for comparison across cultures and healthcare systems. As it is, however, it is likely that people living in high-service environments will appear to be aging more rapidly than those with poorer access to healthcare. But we know that this is not because those with worse access are aging more rapidly on the whole.

I found the literature review somewhat limited, and potentially problematically so, since there are a proliferation of these types of efforts. It would be useful to know, for example, how this paper expands information available from other biological age calculators. For example, GrimAge (Lu et al. 2019) is an epigenetic clock that has recently emerged that is quite interesting, but perhaps more compelling to this paper is its correspondence/deviation from the Cambridge Multimorbidity Score, which seems to be calculated in a similar overall way but appears to be better validated (Payne et al. 2020).

We thank the reviewer for raising important points. We added these references that are relevant to this manuscript.

We appreciate the GrimeAge metric. However, this metric is tuned to cardiovascular risk factors and includes age, with smoking as an environmental factor. Our goal has been to develop health metrics that capture health in various dimensions and that capture the rate of aging at any age and can be used to disentangle mechanisms of diseases, aging and precision approaches. We added more dimensions of the health metrics that show bodily systems age heterogeneously, not synchronously, and have used the Body Clock to explore other health dimensions such as Speed-Body Clock and Disability Body Clock. While GrimAge is a biomarker developed based on PAI1, a strong marker of cardiovascular disease, and smoking as an environmental risk factor of diseases, we developed Body Clock that can be used to differentiate functional resilience from Bodily System health and disentangle mechanism of deficits from resilience and can be used to disentangle genetic and epigenetic mechanisms underlying these health metrics. Whether GrimAge tracks better than other biomarkers of aging in those who are not smokers and not experiencing cardiovascular disease and still age, is not clear.

We appreciate the reviewer's concern regarding access to the diagnostic tools and therefore discrepancy in rate of aging due to a lack of information. However, this was not the case in the BLSA and InCHIANTI data, and all participants had equal access to the comprehensive medical diagnosis that have been used in conducting these two studies. Moreover, in the NHANES data, in which most medical information is survey-based using symptoms, medical and medication history, we used the BLSA model to predict BODN and developed the Body Clock which significantly predicted mortality, SPPB and geriatric syndrome in this dataset. That reveals the potentials of using our models in the settings with limited access to health care using surveys and our algorithms data from BLSA to predict their Body Clock and predict cognitive functional decline and disability. The Bayesian approach allows us to update the model and predict new people's health. This metric can be implemented in medical practice and by using symptoms can predict the Body Clock and other Clocks as well as the rate of aging. Indeed, our approach can be used to combat health disparity regarding calculating the rate of aging when we implement in health settings.

In summary, in this study of the BLSA, all individuals went through full medical examinations performed by a trained health professional (Nurse practitioner in the BLSA and in InCHIANTI). Thus, the out-of-data validations of the analyses was performed using BLSA model as the main model and the InCHIANTI and NHANES data as out-of-data prediction validation (**Highlighted manuscript Page 16-17, Supplemental Info Figure S1- a-b**). The results showed that we can use the high-quality data with full medical examinations to create the Body Clock and other metrics in new data collected by survey.

Using biomarkers to address patients' needs requires a similar medical setting as medical examinations. Indeed, developing a globally unified clinically based measures that can capture the rate of aging can help develop valid biomarkers that can show how they are related to diseases. Developed biomarkers based on mortality or only based on chronological age have not been strongly associated with all chronic diseases. For example, the PhenoAge biomarker developed by Levine and colleagues was mainly associated with cardiovascular disease in UK biobank. Moreover, metrics such as GrimAge would be difficult to interpret in terms of health and difficult to make accessible in remote areas with a lack of health systems.

We thank the reviewer for pointing to the Cambridge study. In this study, the authors modelled the association between 37 morbidities and 3 key outcomes (primary care consultations, unplanned hospital admission, death) and constructed a general outcome multimorbidity score by averaging the standardized weights of the separate outcome scores. The authors weighted each disease similar to Multimorbidity-Weighted Index by Wei et al. (PMID: 28605457) so that the diseases received weight if they predicted such outcomes. However, such a weighting system can underestimate disease impact in case of resilience.

In summary, we believe that the Body Clock and the developed eight-dimensional tool (Health Octo Tool) have the potential for tracking health and rate of aging and can be used to better understand mechanism of aging and health and capture heterogeneity of aging at the individual levels.

Minor issues:

The first "c" in the manuscript is typed in bold face.

We thank the reviewer for pointing this out. We corrected the typo.

The "SPPB" is mislabelled in the tables as SBPP.

We thank the reviewer for the point. We corrected the typo.

Firstly, we extend our sincere appreciation to the reviewer for dedicating time to thoroughly review our manuscript and for providing insightful and constructive comments aimed at enhancing its quality. We have diligently addressed each comment and incorporated necessary revisions accordingly.

The authors have mostly addressed my initial comments. The additional data substantially strengthen the paper, adding confidence that the method illustrated generalizes beyond the cohort in which it was developed. However, I think the core analyses establishing clock predictions of important outcomes could be better represented in the figure set, which largely reports details of the clock, but lacks compelling visualization of evidence that the clock matters. I also think that A LOT of work is needed to improve clarity and readability.

We appreciate the suggestion from the reviewer, and in response, we have included Receiver Operating Characteristic (ROC) and Area Under the Curve (AUC) graphs for Body Clock predicting binary outcomes such as SPPB, Geriatric syndrome, disability, and death. The AUC values indicate that Body Clock more strongly predicts these outcomes in both the BLSA and InCHIANTI studies, which are two longitudinal datasets than FI.

In the NHANES dataset, only mortality is regarded as a longitudinal outcome, and we employed Body Clock to forecast mortality. Nevertheless, it's important to acknowledge that the NHANES data marginally underestimates Body Clock due to lacking information on specific diseases such as thyroid or macular degeneration in certain cohorts. Despite this constraint, the AUC for mortality prediction in this cohort remains at 0.8, surpassing the performance of the Frailty Index. These findings have been integrated into the manuscript, and they are emphasized in both the results and discussion sections.

1. I am still missing analysis comparing the new body clock to existing frailty indices. The response letter says they are there. But I had trouble finding them. Maybe they could be reported in an easier to find place? Basically, it would be nice to see IRRs and HRs with their CIs for the various outcomes in the various datasets plotted side by side parallel effect-sizes estimated for a standard frailty index. This type of visualization is a mainstay of work in epigenetic clocks and is tremendously clarifying for readers re the robustness and value added of a new measure.

We developed FI that includes the same diseases, treating disease levels as an item, while excluding late-onset phenotypes. Our aim was to compare the FI approach with Body Clock in predicting binary late-onset outcomes as well as Disability Index (DI) as a continuous outcome combining functional and cognitive disability.

In Bayesian statistics, we employ an approach where each data point for each individual is analyzed across millions of models through iterations and chains using the MCMC method. This process yields the "elpd" (expected log pointwise predictive density), which averages the logarithm of predictive densities for each data point. The elpd serves as a metric for comparing models, indicating how effectively they predict observed or new data points. Higher elpd values signify better predictive performance. Predictive density refers to the probability density function (PDF) or probability mass function (PMF) of observed data given the model.

By comparing elpd values across different models, we can discern which one offers the most accurate predictions for the data. Consequently, we evaluated the model performance of Body Clock and Frailty Index in predicting Disability Index (DI) and detailed the findings in the supplement, now incorporated into the results section of the manuscript. Additionally, to address the reviewer's comment, we examined Receiver Operating Characteristic (ROC) and Area Under the Curve (AUC) of Body Clock and FI score in predicting binary outcomes. Body Clock much stronger predicted all binary outcomes in both BLSA and InCHIANTI data and mortality as future outcome in the NHANES data.

While we reported the HR for Body Clock, we refrained from doing so for FIs due to the risk of misinterpretation, especially with spurious HR values like 3750 in poorly performing models. It's crucial to clarify that comparing IRRs or HRs of different models doesn't necessarily indicate which model performs better. Unfortunately, this has become a common practice, leading to biased interpretations. To accurately compare models, their performance needs assessment. In the frequentist approach, this entails calculating the log-likelihood of each model and then comparing them. In our Bayesian approach, employed in this manuscript, we evaluate model performance comparing ELPD of the models which also uses Leave-One-Out for the models for each data point.

Furthermore, we introduced new graphs illustrating the relationship between FI and Body Clock across all three databases, highlighting their high visual correlation. However, it's of paramount importance to note that while FI and Body Clock exhibit strong visual correlation, Body Clock effectively captures health heterogeneity, as evidenced by instances where identical FI values correspond to varying Body Clock values. While some may suggest categorizing diseases in FI by severity, such categorization is arbitrary. Body Clock, on the other hand, incorporates true disease estimates into the BODN for each individual, derived from millions of models created via iterations and chains, and validated using Markov Chain Monte Carlo (MCMC) sampling.

We believe this comprehensive approach adequately addresses the reviewer's concerns. All relevant graphs have been included in the manuscript and are highlighted in the results and discussion sections.

2. It would also be useful to see how the models of BODN (i.e. the regression weights estimated for different diseases in the model used to derive body clock) compare across the 3 datasets. Are they similar? It would seem to me that if BODN and Body Clock are meant to be *measures*, the approach would be to derive the Body Clock model in one dataset and then apply it in another. However, if instead this suite of tools is intended as a set of *methods* that can be used to discover properties within a dataset, then it would seem a key observation is how those discovered properties may vary or be conserved.

We appreciate the question from the reviewer. As detailed in the statistical method section and further elaborated in the supplement, we applied the same approach consistently across all three databases and ensured that the curated data were as coherent as possible. In the NHANES dataset, certain cohorts lacked information on thyroid status, eye exams, and osteoporosis, while individuals aged over 85 were all recorded as 85 years old. This may lead to an underestimation of Body Clock or Body Age in this data. However, Body Clock still predicted mortality stronger than FIs. We added these results to the manuscript, which are highlighted.

We added this part to the discussion.

For the BLSA dataset, we have provided graphs illustrating the effect estimate of diseases, and we have extensively reported estimates for single disease models, single system models, and whole systems in the Supplement, along with the weights of the models of BLSA and InCHIANTI data. We replicated Body Clock across all three databases, which is the accumulative regression weights estimates of BODN from all diseases for each individual and obtained using post hoc analysis of BODN when all diseases included in one model.

All these details are stated in the manuscript and in the supplement.

3. More generally, the paper could be made more user friendly. There are a lot of data being integrated with some pretty sophisticated methods. If there is real value add to prediction of adverse aging outcomes – or even if this novel approach just matches standard FIs in terms of predictive power -- there is something new and important here. But some additional work is needed to make that contribution available to readers. Below are some suggestions toward moving the paper in this direction:

a. There are SO many new measures in this paper -- Body Clock, BODN, Body Age, Bayesian Disability Index... There are a lot. I found it very hard to keep track of what each one was and why they mattered/what they meant in terms of the overall argument. I think the paper needs a text-box or figure that provides a glossary of these new measures that defines them and also explains their role in the analysis. A graphical-abstract-type figure could also accomplish this. I think figure 1 is suppose to do this job. It doesn't. In fact, the initialization-rich figure and the exceptionally terse legend only serve to the confuse the reader more. Most will be driven to despair (or just the next open browser tab) before they get a chance to appreciate what the authors are after.

We deeply appreciate the reviewer for dedicating time to thoroughly read our manuscript and for providing insightful comments. We have made several improvements in response to the feedback received. Firstly, we have replaced Figure 1 with a new graph illustrating the overall concepts of the Clocks. Additionally, we have included a table in the method section that describes all relevant terminology. To guide readers, we have added a sentence at the beginning of the results section directing them to this table and Graph 1. Furthermore, we have enhanced the clarity of the text by providing more explicit explanations on the use of each clock and their age estimators, all of which have been highlighted in the manuscript.

b. The abstract is impossible to follow. I read it maybe a dozen times and couldn't figure it out. If a reader starts there, they will stay lost forever. In contrast, the opening paragraphs of the introduction are quite clear and make a compelling case for the new measure/suite of measures being introduced. Perhaps redrafting the abstract focusing on the high-level goals of the project would be a better strategy.

We are grateful to the reviewer for their comment. We have restructured and rewritten the entire abstract to articulate the rationale behind the necessity for multiple health metrics.

c. Simpler language would go a long way toward readability. For example, on page

4, still in the introduction, the authors define their first novel measure, BODN, as “the number of organ systems with at least one deviation from health due to diseases or impairments, providing an unequidistant ordinal value so that the severity of diseases or disease levels can variably contribute into this unequidistant metric, and we can obtain their coefficients as the weight of their contribution into BODN using a Bayesian framework”

I have no idea what that means. I have high confidence that neither do 99% of the readers of Nat Aging. From the first paragraph of the results, I would be inclined to understand BODN as simply the count of organ systems with evidence of pathology. But that sentence suggests something much more exotic. And down in the results, after the apparently simple explanation as count of organs with pathology, the text returns to “nonequidistant...” Is the point that BODN is calculated as a count, but modeled as an ordinal variable to relax the assumption that each step up the count represents the same magnitude of increase in body entropy?

We appreciate the reviewer’s feedback.

Although at first glance, Body Organ Dysfunction Number (BODN) appears to be a count of the number of dysfunctions, its nature is actually ordinal. To illustrate, consider a patient with diabetes: initially, it manifests as high blood sugar, progresses to kidney dysfunction as a consequence, and eventually leads to diabetic retinopathy. If encountering a patient with all three conditions, it might be tempting to count them as three deficits, but they are part of one organ. Given the effects of organs on one another, the deficits/diseases of other organs progress. Therefore, in reality, to capture this progress from one organ to another in terms of heterogenous effect of the diseases as we hypothesized, we considered BODN as an ordinal value in the model so that a model incorporating BODN and the effects of diseases on the ordinal BODN can address these complexities. By incorporating disease-specific effects into a model accounting for the individualized effects of each disease predicting BODN, we used post hoc analysis, and obtained Body Clock, which is the predicted BODN given the variable effects of each disease.

We added the points as below to the beginning of the Result which is highlighted:

“...we combined medically defined laboratory and clinical examination of different organ systems, spanning from subclinical to clinically severe states, to compute the Body Organ Disease Number (BODN) as the sum of organ systems where at least one deviation from normal health was detected.

We hypothesized that deviation from health in selected organ systems does not uniformly impact the BODN and given that deficits at organ are a process that diseases variably incorporate into it, we included BODN as an ordinal value in the Bayesian models. We conducted Post hoc analyses to derive values of BODN for each organ system including their diseases in the model and developed Bodily System Specific Clocks that estimated age as the organ age proxy (Bodily System Specific Age) in each individual. Subsequently, we integrated information from multiple organ systems into one model to assess their collective impact on BODN and using post hoc analysis derived the predicted BODN as Body Clock, which is conceptualized as a weighted measure of multimorbidity. Body Clock estimated age, as Body Age as proxy of aging body. Walking Speed is one of the phenotypes of aging. We estimated the effect of Body clock on walking speed, termed Speed-Body Clock. To capture its aging heterogeneity, Speed-Body

Clock estimated age, termed Speed-Body Age. Additionally, we developed a Disability Index (DI) using a Bayesian approach combining information on physical and cognitive function. Predicting DI with the Body Clock allowed us to understand how the organ systems health entropy predicts DI and provided Disability-Body Clock and its age estimator as Disability Body Age—a proxy for the rate of aging that incorporates bodily system health and disability states. The model originally developed in the BLSA was subsequently validated in the InCHIANTI and the NHANES cohorts. Collectively, these metrics comprise the Health Octo Tool, a multidimensional health assessment for use in clinical trials, clinics, and research endeavors.”

d. Being more clear earlier on about what data are used to define BODN would be helpful. There is a lot of talk about modeling BODN and BODN dynamics from disease data. But one wonders how disease data are distinct from the data used to calculate BODN. Is BODN computed from biomarkers? I couldn't find the details.

We appreciate the comments from the reviewer. BODN is defined based on the organ systems in the body, with each system has its own set of diseases. For instance, the cardiovascular system encompasses diseases such as hypertension (is defined using systolic and diastolic blood pressure above the cut-points accepted in Geriatric text books), ischemic heart disease (clinically defined as having ischemic pain and other related symptoms on Electrocardiogram[ECG]), peripheral artery disease (measured by Ankle brachial Index), different arrhythmia(based on ECG), and congestive heart failure (based on signs and symptoms such as shortness of breath, ejection fraction value, and other medical sign and symptoms), each varying in severity as well. We considered an organ system to have morbidity if it exhibited at least one of these diseases or deviation from normal health (subclinical states in some of diseases like sub-clinical hypothyroidism. It's important to note that disease diagnosis in the medical field relies on a combination of factors including medical history, physical examination, organ specific laboratory tests and treatment utilized. Therefore, BODN is derived from these diagnoses, which in turn are based on a variety of clinical information. We have incorporated this explanation into the manuscript and have emphasized its significance.

We added the above explanation to the Supplemental Method.

e. The same goes for body clock. The authors offer very little detail about how these measures are actually constructed, offering a single, ultra-dense sentence that makes no reference to the data being used, and then galloping off to describe the properties of the novel metric. The closer I looked, the less I understood.

We appreciate the reviewer's suggestion. We stated how Body Clock is developed in response to item c. “By incorporating disease-specific effects into the model and predicting BODN as an ordinal value, accounting for the individualized effects of each disease, we can obtain the Body Clock as post hoc analysis, which is the predicted BODN given the variable effects of each disease.” We added thi spart to the manuscript, which is highlighted.

MINOR

The authors should not abbreviate chronological age as “c-age” this is unnecessary and makes the paper harder to read/understand. There are enough new things in this paper that we don't

need new names for things we already understand.

We understand that many terminologies can be cumbersome, and we appreciate the time and deep thinking invested by the reviewer. To distinguish between biological age and chronological age, and considering the length of the manuscript, we have used "c-age" to represent chronological age. We would appreciate it if the reviewer agreed to allow us to keep "c-age" as it is.

REVIEWER COMMENTS

Reviewer #2 (Remarks to the Author): The authors have done a lot of work and substantially addressed one of my two major concerns. In my last review I flagged needs for (1) comparative validation vs. frailty indices; and (2) clearer explanation of the method. I think the authors delivered on (1). They clearly made an effort on (2). But, in my view, they are not there yet – frankly, they are not close.

We would like to extend our sincere appreciation for the encouraging feedback on our manuscript. We greatly value your recognition of the comparative validation with the Frailty Index and our efforts to clarify the methodology. Our responses are highlighted in underlined text, and the additions to the manuscript are enclosed in quotation marks. The Reviewer's comments are presented in italic text.

I do not understand what is at the core of this toolkit. I think the idea is that by modeling a count of organ systems with pathology as an ordinal variable from longitudinal data on individual diseases, the authors can generate a predicted value that represents something more sophisticated/nuanced than the original count. Everything else follows from that predicted value. But it's not clear from the paper as written what that predicted value is. It would be helpful (a) to state that clearly how BODN is measured (i.e. from what variables combined in what way taking on what resulting distribution of values); and (b) to do the same for the various models of BODN – i.e. what are the predictor variables? I think the underlying theory of measurement here is that modeling a multimorbidity index from longitudinal organ-system data can generate predicted values that provide a deeper summary of the aging process than the count itself. But honestly I'm just not sure.

We appreciate the reviewer's comments. As the reviewer noted, the values obtained from the post-analysis (post-hoc) prediction of the models are the basis of all clocks and rate of aging metrics.

We have added more detailed explanations on BODN and provided a list of diseases considered for each organ, directing the reader to Table 1 (formerly Table S1). We used combinations of medical examinations (such as bone mineral density from scans), laboratory tests (like thyroid hormones), and questions on medical history and medication, as indicated in Table S1 and Table 1, to define diseases. As explained, BODN represents the number of organ systems with at least one disease. However, since organs are not entirely independent of one another, disease progression from one organ to another does not occur abruptly but rather progressively. This indicates that there are inherently unequal distances between successive values of BODN. Therefore, we consider BODN as an ordinal value with unequal distances. In the Bayesian framework, a cumulative family (or cumulative link model) is used to model ordinal outcomes,

where the response variable has a natural ordering, but the intervals between levels may not be equal.

In Bayesian ordinal regression, the approach to an ordinal outcome using a cumulative function is based on a latent variable. We used concepts from the book Bayesian Data Analysis (reference 21: Bürkner PC, Vuorre M. "Ordinal Regression Models in Psychology: A Tutorial." Advances in Methods and Practices in Psychological Science, 2, 77-101 (2019)) and reference 23 in our manuscript (Gelman A, Carlin J., Stern H., Dunson D., Vehtari A., Rubin D. Bayesian Data Analysis (2014)).

First to clarify the main stem of the Bayesian approach we borrowed the explanation from the mentioned references and added to the Supplemental Method.

“A statistical model is considered Bayesian if it represents uncertainty in both observed and unobserved variables—typically referred to as data and parameters—using probabilities. This is commonly formalized through Bayes' theorem, which defines the posterior distribution $p(\theta)$ of the model parameters θ , given the data y , as the product of the likelihood $p(y|\theta)$ and the prior distribution $p(\theta)$, along with a normalizing constant $p(y)$ (while y is the data, and θ is the model parameter and p is probability).

$$P(\theta|Y) = \frac{P(\theta) \times P(Y|\theta)}{P(Y)}$$

Then we added the below explanation to the Method section of the manuscript page 31.

“Briefly, the cumulative model operates under the assumption that the observed ordinal variable Y is derived from the categorization of an underlying continuous latent variable \tilde{y} . In this framework, there are latent thresholds τ_k (where $1 \leq k \leq K$) that divide the continuous latent variable \tilde{y} into $K+1$ distinct, ordered categories, which correspond to the observed ordered values of Y .

$$Y = K, \tau_{k-1} < \tilde{y} < \tau_k$$

In this formula K is 14 as number of subsequent values of BODN and \tilde{y} is predicted BODN in the model and where there are all diseases of all organs as covariates, the predicted values are called Body Clock.”

Most covariates are diseases with more than two levels, and to capture the maximum effect of diseases with different levels, we use a new function in the Bayesian framework called the monotonic effect.

We employed this concept to our research from the reference number 22 in our manuscript (Burkner P, Charpentier E. Modelling monotonic effects of ordinal predictors in Bayesian

regression models. *Br J Math Stat Psychol* **73**, 420-451 (2020) and it is stated in the Supplemental Method of our manuscript as below:

“Considering that the levels of ordinal predictors might not be equidistant, we used a function called “monotonic effect” implemented in the Bayesian “brms” software package published previously, which assume that the ordinal predictor is a latent continuous variable with a posterior estimate (total beta coefficient for the ordinal variable). When encountering information from the data, a cut-point is introduced using the “simplex function” ζ , so that $\zeta_i \in [0,1]$. Therefore, ζ is the posterior probability of each level, and i is the number of ordinal categories. We can quantify the posterior estimate (beta coefficient) of each disease level through multiplying the total posterior estimate by the proportion of each level ζ_i , e.g., HTN has three levels, the first level is the reference [normal]; the second and the third level proportions are $\zeta\%$ and $1-\zeta\%$, respectively. With the posterior estimate Θ (total beta coefficient) of the latent continuous HTN, the posterior estimate of each HTN level would be $(\Theta * \zeta\%)$ and $(\Theta * 1- \zeta\%)$, respectively.”

These functions are implemented in the brms package in R as well as possible to employ the function in STAN statistical program as we have provided in the GitHub. We added to the end of the Method as:

“The codes for algorithms are presented in the GitHub page <https://github.com/ssalimi/HealthOctoTool>”

We included organ-specific diseases (listed in Table 1) to predict longitudinal BODN. The predicted value from the model provided the bodily system-specific clock (BSC). We used post-hoc analyses to obtain these predicted values. Similarly, we included all diseases across all organs with longitudinal BODN as the outcome. We called the model's predicted value the Body Clock, a more nuanced metric resulting from the incorporation of diseases into longitudinal BODN.

We added the below formula showing the BODN and the covariates in page 15 manuscript.

$$Fit_{BLSA} = bodn \sim mo(Hypertension) + mo(congestiveHeartFailure) +$$
$$mo(IschemicHeartDisease) + mo(Arrhythmia) + mo(Kidney) + mo(Diabetes) +$$
$$mo(Hyperlipidemia) + PrepheralArteryDisease + mo(Stroke) + mo(Anemia) +$$
$$Thrombocytopenia + mo(GastrointestinalDisease) + mo(Liver) + mo(COPD) +$$
$$Asthma + mo(OralHealth) + mo(Hypothyroidism) + Hyperthyroidism +$$
$$mo(OsteoArthritis) + mo(Osteoporosis) + mo(Hearing) + mo(Eye) +$$

$mo(Depression) + mo(sParkinsons) + mo(Cognition) + Cancer + yrs +$
 $(1 + 1|id), data = BLSA)$

$Body\ Clock_{BLSA} = posterior_predict (fit_{BLSA}, data = BLSA)$

The above equation is performed in the InCHIANTI data as replication and the predicted model formula is:

$Body\ Clock_{InCHIANTI} = posterior_predict (fit_{InCHIANTI}, data = InCHIANTI)$

Chronological age regressed over Bodily System Specific Clocks and the Body Clock with the predicted values representing Bodily System Specific Age and Body Age, respectively.

$Fit_{Age} = Chronological\ Age \sim Body\ Clock, data = BLSA ,$

$Body\ Age = postero\ predict(fit_{Age})$

The same formula applied to the InCHIANTI data as replication.”

A similar approach was applied to the Speed-Body Clock, as shown in the formula below. From the Bayesian model, we obtained the predicted value, which we call the Speed-Body Clock, as it reflects the influence of the Body Clock on walking speed.

$Fit_{BLSA} = Walking\ Speed \sim Body\ Clock + height, data = BLSA$

$Speed - Body\ Clock = Postrior\ predict (Fit_{BLSA})$

A similar formula applied to the InCHIANTI data.”

We added this formula to the manuscript page 17.

Not all clinical diseases equally impact walking speed, which we capture through the Speed-Body Clock. Additionally, individuals of the same chronological age may have different rates of aging in terms of how the Body Clock affects walking speed. This heterogeneity in the aging rate is captured by regressing chronological age on Speed-Body Clock.

Speed-Body Age represents the rate of aging in terms of the Speed-Body Clock.

" $Fit_{BLSA} = Age \sim Spreed - Body Clock + Height, data = BLSA$

$Speed - Body Age = psoterior predict (Fit_{BLSA})$

The same formula applied to the InCHIANTI data as replication.”

We added this formula to the manuscript page 18.

To assess the effect of the Body Clock on physical and cognitive disability, we developed the Disability Index using items from the Activities of Daily Living (ADL), Instrumental Activities of Daily Living (iADL), the Mini Mental State (MMS) exam, bowel or urinary incontinence, and falls. We applied a Bayesian approach, where the probability of experiencing an event/item is conditioned on the total number of items measured. This allows us to obtain the predicted probability of events, conditioned on the total trials (total items), as the Disability Index

" $Fit_{BLSA} = probability of (events) | trial (total events) \sim 1 + (1 | id), data = BLSA$

$Disability Index = posterior predict (Fit)$

The same formula applied to the InCHIANTI data as replication.”

These equations are added to the manuscript page 20 and the Bayesian formula has been described in the Supplemental Method as below:

“Zero Inflated Beta Binomial. We utilized the Zero-Inflated Beta Binomial model to create the new Disability Index, Disability-Body Clock, and Disability-Body Age. Initially, a Bayesian-based beta binomial model was developed to calculate the probability of events (in this case, late-onset outcomes) given the number of trials (which represented the total number of disability items, 47 items in this context) using the formula below:

$$P(event|trials, p) = \binom{trials}{event} p^{event} (1 - p)^{Trials-event}$$

$$p_{i=} = \frac{exp(\eta_i)}{1+exp(\eta_i)}$$

p is probability of the event for observation i and η_i is the linear predictor term.

For mixing the zero inflated distribution to the beta binomial the formula is:

$$p(event|trails, \mu\Phi, (1 - \mu)\Phi, \theta)$$

$$= \begin{cases} \theta(1 - \theta) Beta Binomial (event = 0 | trials, \mu\Phi, (1 - \mu)\Phi) \\ (1 - \theta) Beta Binomial (event > 0 | trials, \mu\Phi, (1 - \mu)\Phi) \end{cases}$$

μ : standard Beta Binomial distribution

Φ : densities of values

Θ : probability of zero values for the events

$(1 - \theta)$: Probability of the event values more than zero”

Not everyone with an increased Body Clock experiences disability. To evaluate the impact of the Body Clock on disability, we used the following model. The model's predicted value, referred to as the Disability-Body Clock, captures the individual heterogeneous effect of the Body Clock on disability. We added the below text and equations to the manuscript page 20:

“Not everyone with an elevated Body Clock experiences disability. This heterogeneous process can be captured using the Disability-Body Clock (defined as the sum of positive events conditioned on the total number of measured events and predicted by the Body Clock). This approach enabled us to explore how the Body Clock predicts the model-based physical and cognitive DI based on events within the total 47 measured items (Table S2).

$$Fit_{BLSA} = Disability\ Index \sim Body\ Clock, data = BLSA$$

$$Disability - Body\ Clock = posterior\ predict (Fit)$$

The same formula applied to the InCHIANTI data as replication.”

Furthermore, individuals of different ages can manifest different effects of the Body Clock on disability. To capture this heterogeneity in the rate of aging, we regressed chronological age over Disability-Body Clock. The predicted value of this model is Disability-Body Age.

We added the below text and equation to the manuscript page 20-21

“Additionally, individuals of varying ages may exhibit different effects of the Body Clock on disability. To account for this heterogeneity in such aging rate, we regressed chronological age over the Disability-Body Clock. The predicted value from this model is referred to as Disability-Body Age.

$$Fit_{BLSA} = Age \sim Disability - Body\ Clock, data = BLSA$$

$$Disability - Body\ Age \sim posterior\ predict (Fit)$$

The same formula applied to the InCHIANTI data as replication.” (Page 21).

I also can't tell if the tools reported in this paper are algorithms for which parameters are estimated in the training data (BLSA) and variable values are input from the test data (InCHIANTI, NHANES) or if instead they are methods that can be applied to new datasets to derive dataset-specific parameters. Both can be useful. But I am not clear on which one is being introduced here.

Thank you for your insightful comment. We appreciate the need for further clarification.

Indeed, we have employed both replication and validation approaches. First, for replication, we conducted data-specific analyses on the InCHIANTI dataset using the same method as used for analyzing the BLSA dataset. These data-specific parameters are reported in Supplementary Tables S4a–S4i.

Additionally, we used the parameters from the training data (BLSA) and the input variables from the test data (InCHIANTI or NHANES), as described in our out-of-sample validation on page 24 of the manuscript. We used graphical model performance within a Bayesian framework, which demonstrated that the predicted BODN (Body Clock in the InCHIANTI and NHANES datasets) aligns well with the observed BODN (input data) in these datasets, as presented in the Supplemental Method (Fig. S10a & b). We have also added a graph illustrating the relationship between the Body Clock obtained from data-specific parameters and the Body Clock derived from the test data using BLSA parameters applied to the InCHIANTI dataset as test data.

To avoid confusion, we have further clarified the out-of-sample validation approach and when we used InCHIANTI for replication on page 9 as below:

Validation applying parameters of the BLSA data as training set to the new data as test sets.

“In the Bayesian framework, using parameters from the training data (BLSA) to apply to a new data input assesses out-of-sample predictive accuracy. The parameters of the BLSA entropy model, fitted with BLSA data, were tested by making predictions for individuals in the InCHIANTI study and applying them to the National Health and Nutrition Examination Survey (NHANES) data from 2003 to 2018²⁷. We compared the observed values and predicted Body Clock using visualization function *pp_check* from *brm* package. Additionally, we examined the relationship between the Body Clock derived from the replicated analysis of InCHIANTI to validated version obtained from applying parameters of BLSA data to InCHIANTI data.”

and in page 24 as:

“Applying the BLSA parameters (training set) to the NHANES and InCHIANTI data (validation sets) to develop the Body Clock

In the out-of-data the validation is the same concept as using training data parameters (BLSA model) to a new data (InCHIANTI or NHANES) as test sets to obtain Body Clock. We used the BLSA full entropy model to predict out-of-data input in the NHANES and InCHIANTI data (test sets). The predicted and observed values in both studies were matched as depicted using posterior predicted check in Bayesian framework (Fig. S10 a&b). There is a strong correlation between the Body Clock derived directly from the analysis of the InCHIANTI dataset (as a replication) and the Body Clock obtained by applying BLSA parameters to the InCHIANTI dataset (as validation set) (Fig.6e).

$Fit_{BLSA} = BODN \sim mo(Disease1) \dots + mo(DiseasesN) + time + (1|id), data = BLSA$

Predict in InCHIANTI (test set) = posterior predict (Fit_BLSA, data = INCHIANTI)

Predict in NHANES (test set) = posterior predict (Fit_BLSA, data = NHANES)

There is a strong correlation between the Body Clock derived directly from the analysis of the InCHIANTI dataset (as a replication) and the Body Clock obtained by applying BLSA parameters to the InCHIANTI dataset (as validation test set) (Fig.6e).

The authors need to be clear about what the measurements they analyze are and what the models they fit to those measurements include as covariates. They should also provide lists of the variables they are including in their analyses (e.g. in supplementary tables if necessary) and equations that show how the parameters they describe are estimated.

In the BLSA, medical diagnoses are determined through a combination of medical examinations, patient history, and systematic health reviews. We adapted these criteria for the NHANES data, incorporating all relevant questions that provide data for the algorithms we developed. The disease-related questions, as referenced, are now included in the manuscript, with a detailed list provided in Table S1 of the Supplemental Methods. The disease definitions and their respective levels are outlined in Table 1.

The items for the Frailty Index are explained as below in page 22:

“The list of diseases used to develop the Frailty Index (FI) is similar to the Body Clock, but instead of considering different levels of disease severity, each condition is treated as a separate binary variable. For example, in the case of ischemic heart disease, angina pectoris and acute myocardial infarction are considered distinct deficits in the FI, whereas the Body Clock treats them as different levels of the same condition. Similarly, for eye diseases, the Body Clock treats cataract, glaucoma, and macular degeneration as varying levels of eye disease, while in the FI, each condition is treated as a separate binary variable (e.g., having cataract vs. no cataract). Overall, the same diseases were considered in both the FI and the Body Clock. The primary difference lies in the method used to develop these health metrics (Table 1).”

Finally, related to the above comment, this paper is presented as introducing a toolkit. But I can't tell what the tools are. Are there equations published in the paper that constitute the tools? (I didn't see any) Is there software that will be published with the article that will allow others to implement these tools in new data? If so, what inputs does that software require?

In studies of biological clocks predicting age-related phenotypes, typically single chronic diseases or thresholds of multimorbidity (defined as the number of diseases) of less than 2, 2-4, and 5 or more have been used. For the first time, we have used clinical information and a Bayesian approach to represent these clinical phenotypes as clocks and rates of aging, which can be used as outcomes for other biological metrics or in clinical trials. We have now included Bayesian statistical equations and formulas for greater clarity in the manuscript and on GitHub.

We have added the Bayesian statistical equations for ordinal regression, the monotonic effect function, and the Zero Inflated Beta Binomial approach, which are the main Bayesian methods employed in our tools. Similar to other tools (such as Dunedin epigenetic clocks, PhenAge, Grim Age) that apply statistical methods to a set of markers to create their tools, we have applied new Bayesian concepts to clinical data to develop the Health Octo Tool. Additionally, this is the first time system clocks and rates of organ-specific aging (Body System Specific Clock and Age) have been developed.

In this paper, we introduced the concepts, equations, parameters, replication, and validity. A software package is currently under development to make these tools more user-friendly and accessible for further data production in the near future. However, this software is not included in this manuscript. We introduced the concepts, formulas, and code.

The validation analysis is a bit more clear – at least the evidence that the metrics computed in the analysis are predictive of stuff we care about and generally more so than standard frailty indices. But here too things are needlessly opaque. What diseases were counted to compute frailty?

We appreciate the positive feedback from the reviewer regarding the validation analysis and the superior performance of the Body Clock in predicting late-onset outcomes compared to the standard frailty index. We have added this explanation to the manuscript on page 22.

“Body Clock superseded Frailty Index to predict late-onset binary outcomes. We used the same list of diseases described in Table 1 to manually develop disease-based FI scores, which is a common metric used to capture age-related deficits^{11, 39, 40}. Rather than considering different levels of disease severity, each condition is treated as a separate binary variable. For example, in the case of ischemic heart disease, angina pectoris and acute myocardial infarction are considered distinct deficits in the FI, whereas the Body Clock treats them as different levels of the same condition. Similarly, for eye diseases, the Body Clock treats cataract, glaucoma, and macular degeneration as varying levels of eye disease, while in the FI, each condition is treated as a separate binary variable (e.g., having cataract vs. no cataract). Overall, the same diseases were considered in both the FI and the Body Clock. The primary difference lies in the method used to develop these health metrics (Table 1).”

SPECIFIC COMMENTS

Introduction para 3 (p4 para2) needs to be re-written. It's critical motivation for the paper, where the authors aim to problematize existing algorithm-based measurements of aging and highlight areas where the new approach adds value. They need to get it right. Here is what stood out to me in the reading: The first sentence lumps together the Horvath, PhenoAge, GrimAge, and DunedinPACE epigenetic clocks as examples of approaches that use chronological age as a reference to predict risk for adverse health outcomes. As written, the sentence is accurate only about the Horvath clock (which is a rather poor predictor of adverse health outcomes).

The other three clocks (which are much better predictors of adverse health outcomes) were developed using different methods. The model used to develop PhenoAge included chronological age as part of a composite reference phenotype. In the case of GrimAge, chronological age is actually a component of the algorithm (so not part of the reference at all). In the case of DunedinPACE, the reference is a composite measure of rates of change in multiple organ systems over 20y of follow-up and sample used to develop it was all the same chronological age (so again, no chronological age in the reference). ^[17]~~[17]~~The second sentence makes the claim that chronological age was included in "equations to predict" various outcomes "as part of the indices". I'm not sure what this is meant to convey. But it again sounds like the researchers include chronological age in the "biological age" measurement they are using to predict outcomes. What is true of the papers cited is that they covary for chronological age in models predicting adverse health outcomes with the goal of isolating the unique information captured in the biomarker algorithm. That's quite a bit different from what the sentence implies. Also, the papers cited are about different measures than the ones introduced in the previous sentence.

The third sentence makes a claim about heteroskedasticity in prediction errors from biological age algorithms. While no citations are provided, I know this claim tends to be true of so-called "1st-generation" epigenetic clocks. That's a pretty limited subset of the space being covered in this para. It's also not an obvious limitation – we might assume that the oldest people look biologically younger than model expectations because, as suggested by their having lived longer than most of their birth cohort, their rate of aging is actually slower...

Fourth, the final sentence of the para concludes with a broad claim about algorithms developed using survival/mortality as a reference being vulnerable to external validity issues arising from the architecture of mortality varying over time and space. I am sympathetic to the idea being expressed here. But it's basically a non-sequitur in this paragraph. In the first case, the authors have not introduced such clocks previously. So the reader doesn't know what they are talking about. But more importantly, this is a big complicated idea and not one to be passed off in a single sentence supported by a lone ref that is mostly about a different topic. I also note that the validation evidence for

clocks trained to predict mortality (Grim, Pheno) is pretty robust across settings with significant differences in mortality architecture.

In sum, this is a bad paragraph from top to bottom. The authors should scrap it and start over. They need an argument for what their new measures add to the literature. This isn't it. For my taste, the next paragraph is equally confused and hard to follow. There is a good argument to be made for this tool. But it has to be made. I cannot follow what is written in this paragraph and the referencing doesn't help – many of the papers don't seem to be relevant to the claims being made... I would suggest scrapping the discussion of biological age metrics altogether. They are difficult to explain and it's not clear to me that this toolkit is intended as a competitor to them. Rather, as articulated in the opening paragraphs, which I liked, it sounds like the goal here is a more sophisticated version of a multimorbidity index. I would focus the argument for the knowledge gap addressed/value added on that.

We acknowledge the reviewer's thorough explanation and comments. Our paper is focused on clinical phenotypes and laboratory tests that are clinically used to define diseases. This is the first time that multimorbidity has been represented as a clock and rate of aging, and its impact on functional phenotypes and disability has been translated into clocks and rates of aging. Additionally, this is the first time that estimated ages of systems and system-specific clocks have been developed. The goal has not been to compete with any other biological clocks or markers. We agree with the reviewer's points and have removed this paragraph from both the Introduction and Discussion.

MINOR: Use of "c-age" to abbreviate chronological age is unnecessarily hard on the reader. Just spell it out.

We appreciate the reviewer's comment. We used 'c-age' to limit word count, but we have now spelled it out as 'chronological age.'

I have not commented on the discussion because I don't feel I understand what comes before sufficiently well to do so.

We appreciate all the comments that have helped us improve the manuscript and hope that the current version is satisfactory.

We appreciate the valuable comments from the reviewer and have provided our responses below each item. Your feedback has been instrumental in enhancing the quality of our manuscript, and we sincerely appreciate your constructive input.

Sincerely

Salimi et al.

Response to reviewer:

- 1) The spaghetti plots could benefit from having an average overlaid. It looks to me as though several of the relationships indicate the presence of accelerated change in older ages, which I think are based on the monotonicity assumptions of the model, so it would be good to allow some curvature to the line of best fit.**

While we appreciate the suggestions, we chose not to create average-based graphs, as they rely on a frequentist approach (e.g., using the ggplot2 package), whereas our analysis follows a Bayesian approach. Our graphs are designed to capture individualized Body Clocks, the rate of aging, and the heterogeneity of individual data points, which become more pronounced with age. They illustrate increasing inter-individual heterogeneity and accelerated aging in older populations. Since conventional average-based graphs may obscure these critical variations, we opted for individual-based spaghetti graphs to better represent the complexity of aging dynamics.

- 2) I'm not sure i see the benefit of this discussion paragraph: "Using Body Clock regressed over chronological age, we determined rate of entropy age as Body Age. Furthermore, Body Age serves as a key link between health entropy and rate of aging at the individual level." The one that follows two after is also short and similarly unhelpful.**

Thank you for the valuable point. We modified accordingly and wrote as:
"Regressing chronological age over Body Clock, we determined the rate of aging in the whole-body systems, in another word, we obtained the rate of aging in terms of health entropy and called this metric Body Age."

- 3) The abstract shouldn't use the word "effect" and should also contain critical datapoints like accuracy and predictive power that are presented in the manuscript and would help to convince the reader of the manuscript's utility.**

Thank you for your valuable suggestion. We removed the word “effect” and added “The Body Clock supersedes the frailty index and predicts disability, geriatric syndrome, SPPB, and mortality with $\geq 90\%$ accuracy.” To the abstract.

4) The abstract says: "Moreover, aging is a multidimensional process which cannot be captured by a single metric. Therefore, we assessed global health examining disease ... ". This is verbose and should be: ""Since aging is a multidimensional process that cannot be captured by a single metric, we assessed systemic health by examining disease ... "

Thank you for the constructive suggestion we modified the abstract accordingly as below:

“Since aging is a multidimensional process, it cannot be captured by a single metric. Therefore, we assessed global health in longitudinal data by examining disease severities in 13 organs, generating Body Organ Disease Number (BODN), which reflects progressive system morbidities.”

5) The sunburst graphs are in their native forms, but this is unhelpful and looks odd in my version. Could you remove the checkbox with "Legend" next to it and also maybe put all of them on a white background? If possible, to make them bigger, that could also help as they are pretty small on my page and difficult to read.

This graph may not appear as intended because it is designed to be interactive and live online. To fully explore its features, please view it by clicking the provided link. For review purposes, we have included a static hard copy. However, we recommend accessing the interactive version via the link for the best experience.